# Learning-Augmented Search Data Structures

**Chunkai Fu**[*]    **Brandon G. Nguyen**[*]    **Jung Hoon Seo**[*]    **Ryan Zesch**[*]    **Samson Zhou**[†]

## Abstract

We study the integration of machine learning advice to improve upon traditional data structure designed for efficient search queries. Although there has been recent effort in improving the performance of binary search trees using machine learning advice, e.g., Lin et. al. (ICML 2022), the resulting constructions nevertheless suffer from inherent weaknesses of binary search trees, such as complexity of maintaining balance across multiple updates and the inability to handle partially-ordered or high-dimensional datasets. For these reasons, we focus on skip lists and KD trees in this work. Given access to a possibly erroneous oracle that outputs estimated fractional frequencies for search queries on a set of items, we construct skip lists and KD trees that provably provides the optimal expected search time, within nearly a factor of two. In fact, our learning-augmented skip lists and KD trees are still optimal up to a constant factor, even if the oracle is only accurate within a constant factor. We also demonstrate robustness by showing that our data structures achieves an expected search time that is within a constant factor of an oblivious skip list/KD tree construction even when the predictions are arbitrarily incorrect. Finally, we empirically show that our learning-augmented search data structures outperforms their corresponding traditional analogs on both synthetic and real-world datasets.

## 1 Introduction

As efficient data management has become increasingly crucial, the integration of machine learning (ML) has significantly improved the design and performance of traditional algorithms for many big data applications. Kraska et al. (2018) first showed that ML could be incorporated to create data structures that support faster look-up operations while also saving an order-of-magnitude of memory compared to optimized data structures oblivious to such ML heuristics. Subsequently, *learning-augmented algorithms* (Mitzenmacher & Vassilvitskii, 2020) have been shown to achieve provable worst-case guarantees beyond the limitations of oblivious algorithms for a wide range of settings. For example, ML predictions have been utilized to achieve more efficient data structures (Mitzenmacher, 2018; Lin et al., 2022), algorithms with faster runtimes (Dinitz et al., 2021; Chen et al., 2022c; Davies et al., 2023), mechanisms with better accuracy-privacy tradeoffs (Khodak et al., 2023), online algorithms with better performance than information-theoretic limits (Purohit et al., 2018; Gollapudi & Panigrahi, 2019; Lattanzi et al., 2020; Wang et al., 2020; Wei & Zhang, 2020; Bamas et al., 2020; Anand et al., 2020; Almanza et al., 2021; Anand et al., 2021; Im et al., 2021; Lykouris & Vassilvitskii, 2021; Aamand et al., 2022; Anand et al., 2022; Azar et al., 2022; Grigorescu et al., 2022; Khodak et al., 2022; Jiang et al., 2022; Scully et al., 2022; Antoniadis et al., 2023b;a; Shin et al., 2023), streaming algorithms with better accuracy-space tradeoffs (Hsu et al., 2019; Indyk et al., 2019; Jiang et al., 2020; Chen et al., 2022b;a; Li et al., 2023), and polynomial-time algorithms beyond hardness-of-approximation limits, e.g., NP-hardness (Ergun et al., 2022; Nguyen et al., 2023; C. S. et al., 2024).

In this paper, we focus on the consolidation of ML advice to improve data structures for the fundamental problem of searching for elements among a large dataset. For this purpose, tree-based structures stand out as a popular choice among other structures, particularly for their logarithmic average performance. However, these structures often have weaknesses for specific use cases that make them sub-optimal for various applications, which we now discuss.

---

[*]Equal contribution, Texas A&M University, `chunkai369@gmail.com`, `bgn@tamu.edu`, `j.seo0917@gmail.com`, `rzesch@tamu.edu`

[†]Texas A&M University, `samsonzhou@gmail.com`

**Skip lists.** One weakness of tree-based structures is that they need to be balanced for optimal performance, and thus their effectiveness is often closely tied to the order of element insertions. For example, the motivation of Lin et al. (2022) to study learning-augmented binary search trees noted that although previous results already characterized the statically optimal tree if the underlying distribution is known (Knuth, 1971; Mehlhorn, 1977), these methods do not handle dynamic insertion operations. In contrast, skip lists, introduced by Pugh (1990a), maintain balance probabilistically, offering a simpler implementation while delivering substantial speed enhancements (Pugh, 1990b). Skip lists are generally built iteratively in levels. The bottom levels of the skip list is an ordinary-linked list, in which the items of the dataset are organized in order. Each higher level serves to accelerate the search for the lower levels, by storing only a subset of the items in the lower levels, also as an ordered link list. Traditional skip lists are built by promoting each item in a level to a higher level randomly with a fixed probability $p \in (0, 1)$.

Querying for a target element begins at the first element in the highest level and continues by searching along the linked list in the highest level until finding an item whose value is at least that of the target element. If the found item is greater than the target element, the process is repeated after returning to the previous element and dropping to a lower list. It can be shown that the expected number of steps in the search is $\mathcal{O}\left(\frac{1}{p} \log_{1/p} n\right)$ so that $p$ serves as a trade-off parameter between the search time and the storage costs.

In many modern applications, skip lists are used because of their excellent search runtime and their space efficiency. Skip lists are often preferred over binary search trees due to their simplicity of implementation, their support for efficient range query, and their amenability to concurrent processes (Shavit & Lotan, 2000; Lindén & Jonsson, 2013), high efficiency for dynamic datasets (Ge & Zdonik, 2008; Pittard & Tharp, 2010), network routing (Hu et al., 2003; Avin et al., 2020), and real-time analytics (Basin et al., 2020; Zhou et al., 2023). Thus while binary search trees have been a long-standing choice for querying ordered elements, skip lists offer a simpler, more efficient, and in some cases, necessary alternative.

**KD trees.** Another weakness of tree-based data structures is that they generally require the data to obey an absolute ordering. However, in many cases, e.g., geometric applications or multidimensional data, the input points can only be *partially* ordered. Thus in 1975, KD trees, which stand for $k$-dimensional trees, were proposed as a more efficient alternative to binary search trees for searching in higher-dimensional spaces in procedures such as nearest neighbor search or ray tracing for applications in computational geometry or computer vision. A KD tree works by picking a data point and splitting along some spatial dimension to partition the space. This process is repeated until every data point is included in the tree, creating a hierarchical tree structure that enables quick access to specific data points or ranges within the dataset.

**Skewed distributions.** Traditional search data structures treat each element equally when promoting the elements to higher levels. This balancing behavior facilitates good performance in expectation when a query to the skip list is equally likely to be any dataset element. On the other hand, this behavior may limit the performance of the data structure when the incoming queries are from an unbalanced probability distribution.

Real-world applications can feature a diverse range of distribution patterns. One particularly common distribution is the Zipfian distribution, which is a probability distribution that is a discrete counterpart of the continuous Pareto distribution, and is characterized by the principle that a small number of events occur very frequently, while a large number of events occur rarely.

In a Zipfian distribution, the frequency of an event $N(k; \alpha, N)$ is inversely proportional to its rank $k$, raised to the power of $\alpha$ (where $\alpha$ is a positive parameter), in a dataset of $N$ elements. In particular, we have $N(k; \alpha, N) = \frac{1/k^{\alpha}}{\sum_{n=1}^{N}(1/n^{\alpha})}$. The value of $\alpha$ determines the steepness of the distribution so that a smaller $\alpha$ value, i.e., closer to 0, makes it more uniform, while a larger $\alpha$ increases skewness.

Zipfian distributions provide a simple means for understanding phenomena in various fields involving rank and frequency, ranging from linguistics to economics, and from urban studies to information technology. Indeed, they appear in many applications such as word frequencies in natural language (Wang & Wang, 2016; Blocki et al., 2018), city populations (Gabaix, 1999; Vitanov & Ausloos, 2015), biological cellular distributions (Lazzardi et al., 2023), income distribution (Sandmo, 2015), etc.

Unfortunately, although Zipfian distributions are common in practice, their properties are generally not leveraged by traditional search data structures, which are oblivious to any information about the query distributions. To improve this performance bottleneck, we propose the augmentation of traditional skip lists and KD trees with "learned" advice, which (possibly erroneously) informs the data structure in advance about some useful statistics on the incoming queries. Although we model the data structure as having oracle access to the advice, in practice, such advice can often easily be acquired from machine learning heuristics trained for these statistics.

## 1.1 OUR CONTRIBUTIONS

We propose the incorporation of ML advice into the design of skip lists and KD trees to improve upon traditional data structure design. For ease of discussion in this section, we assume the items that may appear either in the data set or the query set can be associated with an integer in $[N] := \{1, \ldots, N\}$, which also in the case of high-dimensional data, may be associated with a $k$-dimensional point. We allow the algorithm access to a possibly erroneous oracle that, for each $i \in [N]$, outputs a quantity $p_i$, which should be interpreted as an estimation for the proportion of search queries that will be made to the data structure for the item $i$. Hence, for each $i \in [N]$, we assume that $p_i \in [0, 1]$ and $p_1 + \ldots + p_n = 1$. Note that these constraints can be easily enforced upon the oracle as a pre-processing step prior to designing the skip list or KD tree data structure. We also assume that the oracle is readily accessible so that there is no cost for each interaction with the oracle. Consequently, we assume the algorithm has access to the predicted frequency $p_i$ by the oracle for all $i \in [N]$. On the other hand, we view a sequence of queries as defining a probability distribution over the set of queries, so that $f_i$ is the true proportion of queries to item $i$, for each $i \in [N]$. Although $f_i$ is the ground truth, our algorithms only have access to $p_i$, which may or may not accurately capture $f_i$.

**Consistency for accurate oracles.** We introduce construction for a learning-augmented skip list and KD trees, which gives expected search time at most $2C + 2 \sum_{i=1}^{n} f_i \cdot \min \left( \log \frac{1}{p_i}, \log n \right)$, for some constant $C > 0$. On the other hand, we show that any skip list or KD tree construction requires an expected search time of at least the entropy $H(f)$ of the probability vector $f$. We recall that the entropy $H(f)$ is defined as $H(f) = \sum_{i=1}^{n} f_i \cdot \log \frac{1}{f_i}$.

Thus, our results indicate that within nearly a factor of two, our learning-augmented search data structures are optimal for any distribution of queries, provided that the oracle is perfectly accurate. Moreover, even if the oracle on each estimated probability $p_i$ is only accurate up to a constant factor, then our learning-augmented search data structures are still optimal, up to a constant factor.

**Implications to Zipfian distributions.** We describe the implications of our results to queries that follow a Zipfian distribution; analogous results hold for other skewed distributions, e.g., the geometric distribution. It is known that if the $r$-th most common query/item has proportion $\frac{z}{r^s}$ for some $s > 1$, then the entropy of the corresponding probability vector is a constant. Consequently, if the set of queries follows a Zipfian distribution and the oracle is approximately accurate within a constant factor, then the expected search time for an item by our search data structures is only a constant, independent of the total number of items, i.e., $\mathcal{O}(1)$. By comparison, a traditional skip list or KD tree will have expected search time $\mathcal{O}(\log n)$.

**Robustness to erroneous oracles.** So far, our discussions have centered around an oracle that either produces estimated probabilities $p_i$ such that $p_i = f_i$ or $p_i$ is within a constant factor of $f_i$. However, in some cases, the machine learning algorithm serving as the oracle can be completely wrong. In particular, a model that is trained on a dataset before a distribution change, e.g., seasonal trends or other temporal shifts, can produce wildly inaccurate predictions. We show that our search data structures are robust to erroneous oracles. Specifically, we show that our algorithms achieve an expected search time that is within a constant factor of an oblivious skip list or KD tree construction when the predictions are incorrect. Therefore, our data structure achieves both consistency, i.e., good algorithmic performance when the oracle is accurate, and robustness, i.e., standard algorithmic performance when the oracle is inaccurate.

**Empirical evaluations.** Finally, we analyze our learning-augmented search data structures list on both synthetic and real-world datasets. Firstly, we compare the performance of traditional skip lists with our learning-augmented skip lists on synthetically generated data following Zipfian distributions with various tail parameters. The dataset is created using four distinct $\alpha$ values ranging from 1.01

to 2, along with a uniform dataset. During the assessment, we query a specified number of $n$ items selectively chosen based on their frequency weights. Our results match our theory, showing that learning-augmented skip lists have faster query times, with an average speed-up factor ranging from 1.33 up to 7.76, depending on the different skewness parameters.

We then consider various datasets for internet traffic data, collected by AOL and by CAIDA, observed over various durations. For each dataset, we split the overall observation time into an early period, which serves as the training set for the oracle, and a later period, which serves as the query set for the skip list. The oracle trained using the IP addresses in the early periods outputs the probability of the appearance of a given node, and then the position of each node is determined.

Our learning-augmented skip list outperforms traditional skip lists with an average speed-up factor of 1.45 for the AOL dataset and 1.63 for the CAIDA dataset. Moreover, the insertion time of our learning-augmented skip list is comparable with that of traditional skip lists on both synthetic and real-world datasets. We also observe that our history-based oracle demonstrates good robustness against temporal change, with little shift in the dominant element set. The adopted datasets show that the set of the top frequent elements does not change much across the time intervals in the datasets used herein.

We similarly perform evaluations on our learning-augmented KD tree data structure. We evaluate our KD tree data structure on Zipfian distributions with various tail parameters, and provide a heatmap of average lookup times for elements. We find that for a large variety of Zipfian parameters, ourt method is able to provides large improvements over traditional KD trees. We perform a similar experiment under Zipfian distributions with added noise, and find our data structure still provides considerable improvements in query time.

We additionally evaluate our method on point cloud samples taken from a 3D model. We bin these samples in space, and create our learning-augmented KD tree on these binned samples. When querying this tree with new binned point samples, we find a decrease in average query time as compared to a traditional KD tree under the same conditions. In addition to this experiment, we provide results on real world datasets of n-grams and neuron activation, and similarly find improvements over traditional KD trees.

**Concurrent and independent work.** We mention that concurrent and independent of our work, Zeynali et al. (2024) used similar techniques to achieve the same guarantees on the performance of learning-augmented skip lists that are robust to erroneous predictions. However, they do not show optimality for their learning-augmented skip lists and arguably perform less exhaustive empirical evaluations. They also do not consider KD trees at all, which forms a significant portion of our contribution, both theoretically and empirically.

**Comparison to Lin et al. (2022).** Our work was largely inspired by Lin et al. (2022), who observed that classical literature characterizing statically optimal binary search trees (Knuth, 1971; Mehlhorn, 1977) no longer apply in the dynamic setting, as elements arrive iteratively over time. Thus, they designed the construction of dynamic learning-augmented binary search trees (BSTs). Their analysis for the expected search time utilized the notion of pivots within their trees and thus were somewhat specialized to BSTs. Therefore, Lin et al. (2022) explicitly listed skip trees and advanced tree data structures as interesting open directions. Qualitatively, our results are similar to Lin et al. (2022), as are those of Zeynali et al. (2024). This is not quite altogether surprising because the main difference between these data structures is not necessarily the search time, but the either the ease of construction in the setting of skip lists, or the ability to handle multi-dimensional data in the setting of KD trees.

## 2 LEARNING-AUGMENTED SKIP LISTS

In this section, we describe our construction for a learning-augmented skip list and show various consistency properties of the data structure. In particular, we show that up to a factor of two, our algorithm is optimal, given a perfect oracle. More realistically, if the oracle provides a constant-factor approximation to the probabilities of each element, our algorithm is still optimal up to a constant factor.

We first describe our learning-augmented skip list, which utilizes predictions $p_i$ for each item $i \in [n]$, from an oracle. Similar to a traditional skip list, the bottom level of our skip list is an ordinary-linked

list that contains the sorted items of the dataset. As before, the purpose of each higher level is to accelerate the search for an item, but the process for promoting an item from a lower level to a higher level now utilizes the predictions. Whereas traditional skip lists promote each item in a level to a higher level randomly with a fixed probability $p \in (0, 1)$, we automatically promote the item $i$ to a level $\ell$ if its predicted frequency $p_i$ satisfies $p_i \geq \frac{2^{\ell}-1}{n}$. Otherwise, we promote the item with probability $\frac{1}{2}$. This adaptation ensures that items with high predicted query frequencies will be promoted to higher levels of the skip list and thus be more likely to be found quickly.

It is worth addressing a number of other natural approaches and their shortcomings. For example, one natural approach would be to use the "median" frequency across the items as a threshold to promote elements to higher levels. However, this promotion scheme is not ideal because computing the median frequency at each time would either require an additional data structure for fast update time or increase the insertion time. A potential approach to resolve this issue would be to use a separate threshold probability is set for each level so that only nodes with a probability higher than the corresponding threshold are promoted to the next level. However, this approach seems to result in an unnecessarily large number of created levels if some item appears with small probability, e.g., $\frac{1}{2^n}$. We can thus first filters out the low-frequency elements and place them remain on the bottom level of the skip list and then proceed with using a separate threshold probability for each level. Unfortunately, this approach utterly fails to even match the search time performance of oblivious skip lists when the distribution is uniform, because all items will be in the same level, resulting in an expected search time of $\Omega(n)$. Hence, we ensure that each element still has a chance of being promoted to higher levels even when their probability is less than the corresponding threshold.

We again emphasize that due to the dynamic nature of the updates, existing results on statically optimal binary search trees (Knuth, 1971; Mehlhorn, 1977) do not apply, as observed by Lin et al. (2022). We give the full details in Algorithm 1. For the sake of presentation, we focus on the setting where the queries are made to items in the dataset. However, we remark that our results generalize to the setting where queries can be made on the search space rather than the items in the dataset, provided the oracle is also appropriately adjusted to estimate the query distribution, using the approach we describe in Section 3.

---

**Algorithm 1** Learning-augmented skip list

---

**Require:** Predicted frequencies $p_1, \ldots, p_n$ for each item in $[n]$
**Ensure:** Learning-augmented skip list
 1: Insert all items at level 0
 2: **for** each $\ell$ **do**
 3:     **if** there are no items at level $\ell - 1$ **then**
 4:         **return** the skip list
 5:     **else**
 6:         **for** each $i \in [n]$ **do**
 7:             **if** predicted frequency $p_i \geq \frac{2^{\ell-1}}{n}$ **then**
 8:                 Insert $i$ into level $\ell$
 9:             **else if** $i$ is in level $\ell - 1$ **then**
10:                 Insert $i$ into level $\ell$ with probability $\frac{1}{2}$

---

We first show an upper bound on the expected search time of our learning-augmented skip-list.

**Theorem 2.1.** *For each $i \in [n]$, let $f_i$ and $p_i$ be the proportion of true and predicted queries to item $i$. Then with probability at least $0.99$ over the randomness of the construction of the skip list, the expected search time over the choice of queries at most $20 + 2 \sum_{i=1}^{n} f_i \cdot \min\left(\log \frac{1}{p_i}, \log n\right)$.*

To achieve Theorem 2.1, we first show that each item $i \in [n]$ must be contained at some level $\max(0, 1 + \lfloor \log(np_i) \rfloor)$, depending on the predicted frequency $p_i$ of the item. We also show that with high probability, the total number of levels in the skip list is at most $\mathcal{O}(\log n)$. This allows us to upper bound the expected search time for item $i$ by at most $2C + 2\min\left(\log \frac{1}{p_i}, \log n\right)$. We can then analyze the expected search time across the true probability distribution $f_i$. Putting these steps together, we obtain Theorem 2.1.

We also prove a lower bound on the expected search time of an item drawn from a probability distribution $f$ for *any* skip list.

**Theorem 2.2.** *Given a random variable $X \in [n]$ so that $X = i$ with probability $f_i$, let $T(X)$ denote the search time for $X$ in a skip list. Then $\mathbb{E}[T(x)] \geq H(f)$, where $H(f)$ is the entropy of $f$.*

Theorem 2.2 uses standard entropy arguments that have been previously used to lower bound the optimal constructions of data structures such as Huffman codes. We next upper bound the entropy of a probability vector that satisfies a Zipfian distribution with parameter $s$.

**Lemma 2.3.** *Let $s, z > 0$ be fixed constants and let $f$ be a frequency vector such that $f_i = \frac{z}{i^s}$ for all $i \in [n]$. If $s > 1$, then $H(f) = \mathcal{O}(1)$ and otherwise if $s \leq 1$, then $H(f) \leq \log n$.*

By Theorem 2.1 and Lemma 2.3, we thus have the following corollary for the expected search time of our learning-augmented skip list on a set of search queries that follows a Zipfian distribution.

**Corollary 2.4.** *With high probability, the expected search time on a set of queries that follows a Zipfian distribution with exponent $s$ is at most $\mathcal{O}(1)$ for $s > 1$ and $\mathcal{O}(\log n)$ for $s \leq 1$.*

Next, we show that our learning-augmented skip list construction is robust to somewhat inaccurate oracles. Let $f$ be the true-scaled frequency vector so that for each $i \in [n]$, $f_i$ is the probability that a random query corresponds to $i$. Let $p$ be the predicted frequency vector, so that for each $i \in [n]$, $p_i$ is the predicted probability that a random query corresponds to $i$. For $\alpha, \beta \in (0, 1)$, we call an oracle $(\alpha, \beta)$-noisy if for all $i \in [n]$, we have $p_i \geq \alpha \cdot f_i - \beta$. Then we have the following guarantees for an $(\alpha, \beta)$-noisy oracle:

**Lemma 2.5.** *Let $\alpha$ be a constant and $\beta < \frac{\alpha}{4n}$. A learning-augmented skip list with a set of $(\alpha, \beta)$-noisy predictions has performance that matches that of a learning-augmented learned with a perfect oracle, up to an additive constant.*

To achieve Lemma 2.5, we parameterize our analysis in Theorem 2.1. Due to the guarantees of the $(\alpha, \beta)$-noisy oracles, we can write $p_i \geq \frac{\alpha}{2} \cdot f_i$, which allows us to express the search time $\log \frac{1}{p_i}$ in terms of the true entropy of the distribution and a small additive constant that stems from $\log \frac{1}{\alpha}$. In fact, we remark that even when the predictions are arbitrarily inaccurate, our learning-augmented skip list still has expected query time $\mathcal{O}(\log n)$, since the total number of levels is at most $\mathcal{O}(\log n)$ with high probability. Since the expected query list of an oblivious skip list is also $\mathcal{O}(\log n)$, then the expected query time of our learning-augmented skip list is within a constant multiplicative factor, even with arbitrarily poor predictions.

## 3 LEARNING-AUGMENTED KD TREES

In this section we present details on our novel approach to KD tree construction. First, we present the algorithm that constructs a learning-augmented KD tree. We focus on the setting where queries can be made on the search space rather than the items in the dataset, which is much more interesting for high-dimensional datasets, since even building a balanced tree on the search space could result in prohibitively high query time, as the height of the tree would already be at least the dimension $d$. Nevertheless, assuming that we have a query probability prediction $p_i$ for element $i$ of our dataset, the intuition of our method is straightforward. Whereas a learning-augmented binary search tree would attempt to find a value such that the probability of a query being on either branch of the tree is balanced, high-dimensional datasets do not have an absolute ordering.

Thus, instead of relying on standard techniques to determine the splitting point of our dataset, we find a specific dimension in which there exists a balanced split such that the probability of a query being on either branch of the tree is balanced. However, there can still be high frequency queries that are not in the dataset, which can cause significantly high query time if not optimized. Hence, we also add to the tree construction high frequency queries that are not data points, in order to reject these negative queries more quickly.

We prove the following guarantees on the performance of our learning-augmented KD tree, first assuming that our oracle is perfect.

**Theorem 3.1.** *Suppose $[\Delta]^d$ is the space of possible input points and queries. Let $N = \Delta^d$ and $p_i$ be the probability that a random query is made to $i \in [N]$, given the natural mapping between*

---

**Algorithm 2** Learning-augmented KD tree construction

---

1: **function** BUILDNODE($x$)
2:     $T \leftarrow \emptyset$
3:     **if** $|x| = 1$ **then**
4:         $T = x$
5:         **return** $T$
6:     best $\leftarrow \emptyset$
7:     **for** each dimension $i$ of $x$ **do**
8:         **for** each element $x$ **do**
9:             Compute the probability of points to the left of $x$ on axis $i$
10:             If the probability is closer to $0.5$ than best, update best
11:     $T.\text{axis} = \text{best.axis}$
12:     $T.\text{left} = \text{BUILDNODE}(x : x[\text{axis}] \leq \text{best.value})$
13:     $T.\text{right} = \text{BUILDNODE}(x : x[\text{axis}] > \text{best.value})$
14:     **return** $T$

---

**Algorithm 3** Learning-augmented KD tree

---

1: **function** BUILD(dataset, queries)
2:     $\text{dataset}_f \leftarrow \{x \in \text{dataset} \mid x.\text{prob} > \frac{1}{n^2}\}$
3:     $\text{queries}_f \leftarrow \{x \in \text{queries} \mid x.\text{prob} > \frac{1}{n^2}\}$
4:     $T \leftarrow \text{BUILDNODE}(\text{dataset}_f \cup \text{queries}_f)$
5:     Insert $\{x \in \text{dataset} \mid x.\text{prob} \leq \frac{1}{n^2}\}$ into $T$ using standard balanced KD tree construction
6:     **return** $T$

---

$[N]$ and $[\Delta]^d$. Let $p = (p_1, \ldots, p_N) \in \mathbb{R}^N$ be the probability vector and $H(p)$ be its entropy. Then given a set of $n$ input points, the expected query time for the tree $T$ created by Algorithm 3 is $\mathcal{O}\left(\min(H(p), \log n)\right)$.

The analysis of Theorem 3.1 corresponding to our learning-augmented KD tree follows from a similar structure as the proof of Theorem 2.1. However, the crucial difference is that the universe size is now $N$, which is exponential in $d$. Thus constructions that consider distributions over all of $[N]$ may suffer $\mathcal{O}(\log N) = \mathcal{O}(d \log \Delta)$ query time, which can be prohibitively expensive for large $d$, e.g., high-dimensional data. Hence, our algorithm requires a bit more care in the truncation of queries with low probability and instead, we build a balanced KD tree for any item with less than $\frac{1}{n^2}$ probability of being queried, so that each of their query times is at most $\mathcal{O}(\log n)$. We further remark this implies robustness of our data structure to arbitrarily poor predictions, by a similar argument as in Section 2.

We next prove a lower bound on the expected search time of an item drawn from a probability distribution $f$ for *any* KD tree.

**Theorem 3.2.** *Given a random variable $X \in [n]$ so that $X = i$ with probability $f_i$, let $D(X)$ denote the depth for $X$ in a learning-augmented KD tree. Then $\mathbb{E}[D(X)] \geq H(f)$, where $H(f)$ is the entropy of $f$.*

By Theorem 3.1 and Lemma 2.3, we thus have the following corollary for the expected query time on our learning-augmented KD tree on a set of search queries that follows a Zipfian distribution.

**Corollary 3.3.** *With high probability, the expected query time on a set of queries that follows a Zipfian distribution with exponent $s$ is at most $\mathcal{O}(1)$ for $s > 1$ and $\mathcal{O}(\log n)$ for $s \leq 1$.*

Finally, we show near-optimality when given imperfect predictions from a $(\alpha, \beta)$-noisy oracle:

**Lemma 3.4.** *Let $\alpha$ be a constant and let $\beta \leq \frac{\alpha}{n^2}$. Then the query time for our learning-augmented KD tree with $(\alpha, \beta)$-noisy prediction matches the performance of a learning-augmented KD tree constructed using a perfect oracle up to an additive constant.*

## 4 EMPIRICAL EVALUATIONS

In this section, we describe a number of empirical evaluations demonstrating the efficiency of our learning-augmented search data structures on both synthetic and real-world datasets. We provide additional experiments in Appendix D.

**Skip lists on CAIDA dataset.** In the CAIDA datasets (CAIDA, 2016), the receiver IP addresses from one minute of the internet flow data are extracted for testing, which contains over 650k unique IP addresses of the 30 million queries. Given that the log-log plot of the frequency of all nodes in the CAIDA datasets follows approximately a straight line in Figure 1, the CAIDA datasets can be approximately characterized by an $\alpha$ factor of 1.37. The insertion time is similar between classic and augmented skip lists, while Figure 2 shows that query time is almost halved when using the learning augmented skip lists at different query sizes. These results assume that the predicted frequency of all items in the query stream is accurate, i.e., the probability vector that is used to build the skip list matches exactly the query stream. The speed-up between the query times of the largest learning-augmented and the oblivious skip lists in Figure 2 is roughly $1.86\times$, which is surprisingly and perhaps coincidentally close to our theoretical speed-up of roughly $1.81\times$ on a Zipfian dataset with exponent 1.37.

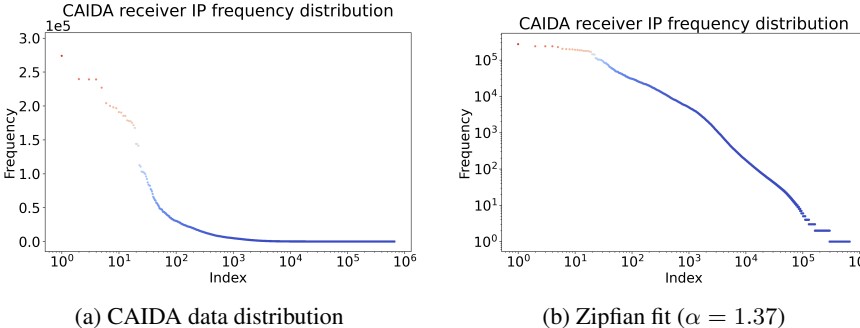

(a) CAIDA data distribution          (b) Zipfian fit ($\alpha = 1.37$)

Figure 1: CAIDA datasets distribution characterization in Figure 1a. The nearly straight-fitted curve in Figure 1b implies that a Zipfian distribution with $\alpha = 1.37$ is a good fit to the CAIDA dataset distribution.

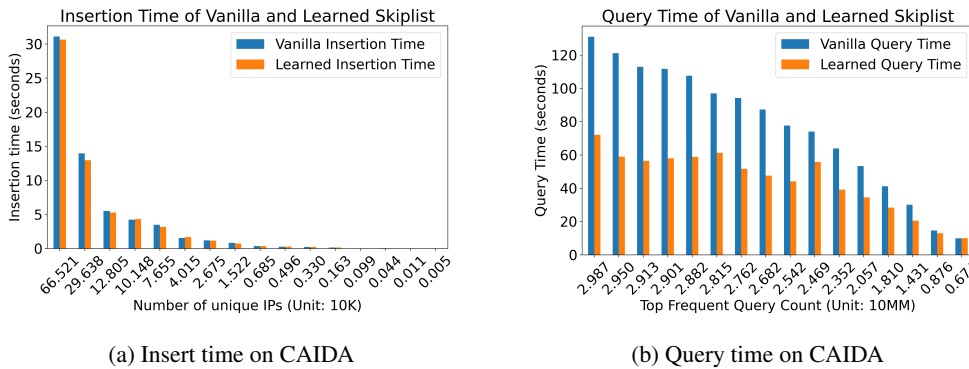

(a) Insert time on CAIDA          (b) Query time on CAIDA

Figure 2: Comparison of insertion and query time on CAIDA for classic and learning-augmented skip lists. This figure compares the insertion and query times under varying numbers of top frequently accessed unique IPs between classic and augmented implementations. The horizontal axis in the two subfigures depicts the same scheme of IP selection, represented in two different ways, e.g., the top 29.9 million queries contain 665210 unique IPs, the next 29.5 million queries comprise 296384 unique IPs, etc.

Next, we demonstrate that our proposed algorithm still manages to outperform the classic skip list even when temporal change exists in the probability vector by comparing the query time for the same set of query elements with different probability vectors being used to guide the building of the structure. For the skip list augmented by a noisy probability vector, the probability vector of elements

during a period of T1 is used as the predicted frequencies. The skip list being augmented by this probability vector has its own set of elements to be organized into the target skip list. Suppose the historic data from T1 contains a set of elements S1, and some future query stream contains a set of elements S2. For each element in our target set S2, if the element is present in S1, then the occurrence probability of this element from S1 will be used to build S2; otherwise, if the element has not shown up during T1 (i.e., in S1), then we assume its probability to be 0. After this, the probability vector is normalized to sum to 1, resulting in a predicted probability vector to be used to build a skip list based on the historic element frequency. Since there is temporal changes in the frequency of elements being queried, the predicted probability vector will show a discrepancy with the true probability vector. The results are presented in Figure 3a and we show that even when the prediction is not perfect, the augmented skip list still performs better than a conventional skip list.

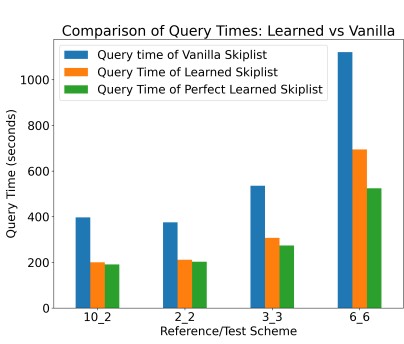
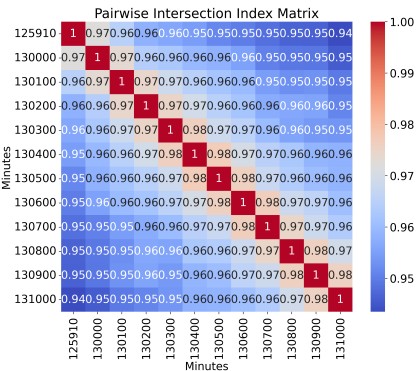

(a) Robustness test on CAIDA datasets  (b) Oracle credibility on CAIDA datasets

Figure 3: Robustness of our learning-augmented skip list to erroneous oracles. In Figure 3b, the labels on the axis indicate the time stamp that the internet trace data is collected, e.g., 130100 means the collection starts at 13:01:00 and lasts for 1 minute.

Figure 3a shows that the skip list with perfect learning shows the best performance, while the skip list augmented with noisy learning performs very close to the scenario with perfect predictions. Moreover, the closer the test data is to the reference data chronologically, the closer the noisy-augmented skip list will perform to the perfect learning skip list. The CAIDA datasets used in this study contain 12 minutes of internet flow data, which totals around 444 million queries. The indices on the x-axis in Figure 3a means:

- 10_2: the first 10 minutes of data are used to create the reference (i.e., oracle) and the last 2 minutes are used to build and test the total query time using the former as reference.
- 2_2: the 9th and 10th minutes data is used as reference and the last 2 minutes are used for testing.
- 3_3: the 1st, 2nd and 3rd minutes of data are used to create reference and the 4th, 5th and 6th minutes of data are used for testing.
- 6_6: the first 6 minutes are used to create the reference and the last 6 minutes are used for testing.

Further analysis of the temporal change of item frequency shows the reason behind the good performance of the history-based oracle. Figure 3b shows the change of intersection index between any 2 given minutes among the 12 minutes of CAIDA data. The intersection index is defined as the ratio of the number of shared queries to the total number of queries of any given 2 minutes of queries. Figure 3b shows that the number of intersects queries has decreased by about 6% after 12 minutes, which indicates that the probability of the majority of the elements will be predicted with good accuracy, resulting in good oracle performance.

**KD trees on synthetic datasets.** KD Trees are commonly used in the field of computer graphics, with applications in collision detection, ray-tracing, and reconstruction. We first generate datasets of $2^{12}$ points in 3-dimensional space, with frequencies given by a fixed Zipfian distribution with parameters $a = 5, b = 2$ – parameters at which our method greatly outperforms a standard KD tree. In order to simulate constructing the tree on noisy data, we multiply the ground truth query probabilities by

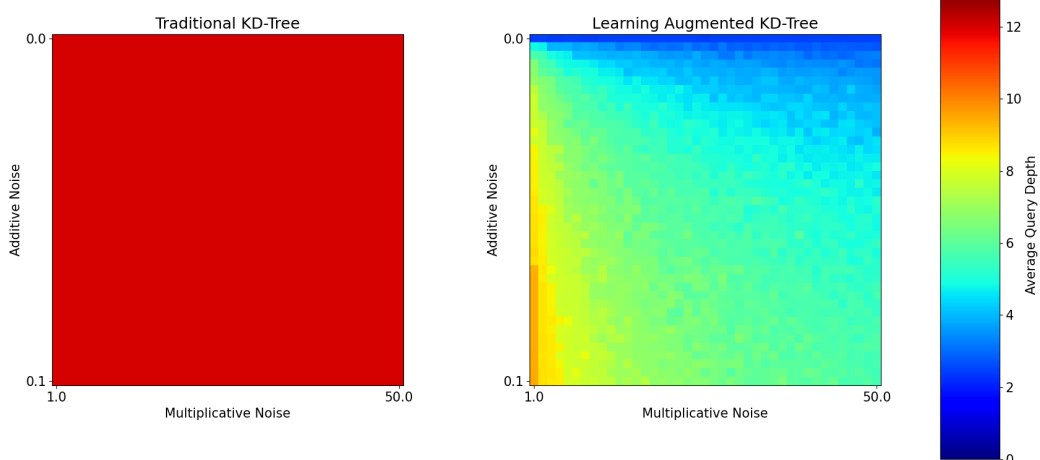

Figure 4: Query time comparison for standard and learning-augmented KD trees with various noise.

numbers sampled uniformly from 1 to $M$, and then add numbers uniformly sampled from 0 to $A$, before renormalizing to form a valid probability distribution. We query the tree $2^{14}$ times, with point queries selected by the ground truth Zipfian distribution. We repeat this process 32 times, and report the median of the average query depth across all runs in Figure 4. We find that our method continues to outperform traditional KD trees under moderate amounts of noise, and at worst, performs on-par with a traditional KD tree.

Next, we generate datasets of $2^{12}$ points in 3-dimensional space, with frequencies given by a Zipfian distribution with parameters $a, b$. In the left plot, we assign these Zipfian weights randomly. In the right plot, however, we assign Zipfian weights with ranks decreasing with the distance to some random data point. We then query the tree $2^{14}$ times, with point queries selected by the same Zipfian distribution. We repeat this process 32 times, and report the median of the average query depth across all runs. We find that, when points frequencies are distributed smoothly over space, our method's performance increases on less skew distributions, as seen in this Figure 5.

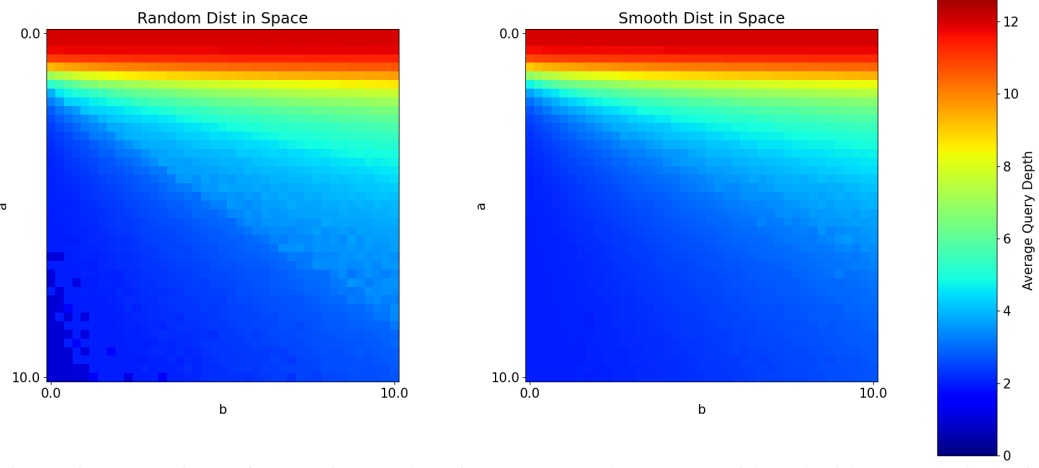

Figure 5: Comparison of query time on learning-augmented KD trees with and without smooth spatial distribution across various Zipfian parameters

**KD trees on 3D point-cloud datasets.** Finally, we evaluate our method on point cloud data generated from the Stanford Lucy mesh (Stan), with dimensions $\sim 1000 \times 500 \times 1500$. We first uniformly sample $2^{22}$ points along the mesh surface, and bin points with resolution 10, and assign lookup frequencies by the number of bin occupants. This results in 32k bins. Note, the resulting frequency distribution for binned cells is not highly skewed.

We then generate a new set of $2^{16}$ surface samples on the mesh, binning them and assigning frequencies in the same way. When looking up with the new samples, our method yields an average query depth of 15.1, while a traditional KD tree yields an average lookup depth of 17.6.

ACKNOWLEDGEMENTS

Samson Zhou is supported in part by NSF CCF-2335411. The work was conducted in part while Samson Zhou was visiting the Simons Institute for the Theory of Computing as part of the Sublinear Algorithms program.

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

## A  ADDITIONAL RELATED WORKS

In this section, we discuss a number of related works in addition to those mentioned in Section 1. This paper builds upon the increasing body of research in learning-augmented algorithms, data-driven algorithms, and algorithms with predictions. For example, learning-augmented algorithms have been applied to a number of problems in the online setting, where the input arrives sequentially and the goal is to achieve algorithmic performance competitive with the best solution in hindsight, i.e., an algorithm that has the complete input on hand. Among the applications in the online model, learning-augmented algorithms have been developed for ski rental problem and job scheduling (Purohit et al., 2018), caching (Lykouris & Vassilvitskii, 2021), and matching (Antoniadis et al., 2023b). Learning-augmented algorithms have also been used to improve the performance of specific data structures such as Bloom filters (Mitzenmacher, 2018), index structures (Kraska et al., 2018), CountMin and CountSketch (Hsu et al., 2019). Specifically, (Kraska et al., 2018) proposes substituting B-Trees (or other index structures) with trained models for querying databases. In their approach, rather than traversing the B-Tree to locate a record, they use a neural network to directly identify its position. Our work differs in that we retain the desired data structures, i.e., skip lists and kd trees, and focus on optimizing their structures to enable faster queries, which allows us to continue supporting standard operations specific to the data structures such as traversal, order statistics, merging, and joining, among others. Our work uses the frequency estimation oracle trained in (Hsu et al., 2019) on the AOL search query dataset and the CAIDA IP traffic monitoring dataset.

Perhaps the works most closely related to ours in the area of learning-augmented algorithms are those of (Lin et al., 2022; Cao et al., 2023; Zeynali et al., 2024). Lin et al. (2022) noted that traditional theory on statically optimal binary search trees (Knuth, 1971; Mehlhorn, 1977) is no longer applicable in dynamic settings, where elements are added incrementally over time. Hence, they developed learning-augmented binary search trees (BSTs) and showed that their expected search time is near-optimal. Cao et al. (2023) then extended these techniques to general search trees, allowing for nodes with more than two children. Cao et al. (2023) also studied the setting where the predictions may be updated, while ultimately still utilizing a data structure that requires rebalancing as data is dynamically changing. Zeynali et al. (2024) also consider the performance of learning-augmented skip lists that are robust to erroneous predictions; we elaborate more on the differences from Zeynali et al. (2024) in Section 1.1. We also note that none of these works consider KD trees at all, which is an important data structure with applications in computer vision and computational geometry, thus forming a basis of our work. For a more comprehensive source of related works in learning-augmented algorithms, see https://algorithms-with-predictions.github.io/.

Beyond the context of learning-augmented algorithms, there is a large body of works that study design of data structures that are optimal for their inputs. For example, while standard binary search trees use $\mathcal{O}(\log n)$ query time, optimal static trees can be constructed using dynamic programming or efficient greedy algorithms (Mehlhorn, 1977; Yao, 1982; Karpinski et al., 1996), given access frequencies. However, the computational cost of these methods often exceeds the cost of directly querying the tree. As a result, a key objective is to construct a tree whose cost is within a constant factor of the entropy of the data. Several approaches have achieved this either for worst-case data (Mehlhorn, 1977) or when the input follows particular distributions (Allen & Munro, 1978).

More recent works have considered using results from learning theory to estimate the query frequencies, rather than assuming explicit access to their values. For example, Cayton & Dasgupta (2007)

studied how to obtain such an oracle for learning-augmented data structures. In particular, they study generalization bounds in the context of learning theory, analyzing the number of samples from an underlying distribution necessary to produce an oracle with a small error rate. On the other hand, Ailon et al. (2011) studied algorithms for sorting and clustering that can improve their expected performance given access to multiple instances sampled from a fixed distribution. Although the high-level goal of improving algorithmic performance using auxiliary information is the same as ours, the specifics of the paper seem quite different than ours, as the paper focuses on techniques for sorting and clustering. Similarly, Ciriani et al. (2002) considers self-adjusting data structures, including skip lists, which can dynamically change as the sequence of queries arrive. However, their methods are catered specifically to the setting where there is access to the queries, whereas our data structures must be constructed without such access and must therefore be able to handle erroneous predictions. Finally, we remark that utilizing techniques for learning theory, there are standard results about the learnability of oracles for the purposes of learning-augmented algorithms (Ergun et al., 2022; Izzo et al., 2021).

## B MISSING PROOFS FROM SECTION 2

In this section, we give the missing proofs from Section 2.

### B.1 EXPECTED SEARCH TIME

We first show that each item is promoted to a higher level with probability at least $\frac{1}{2}$.

**Lemma B.1.** *For each item $i \in [n]$ at level $\ell$, the probability that $i$ is in level $\ell + 1$ is at least $\frac{1}{2}$.*

*Proof.* Note that if $p_i \geq \frac{2^\ell}{n}$, then $i$ will be placed in level $\ell + 1$. Otherwise, conditioned on the item $i \in [n]$ being at level $\ell$, then Algorithm 1 places $i$ at level $\ell + 1$ with probability $\frac{1}{2}$. Thus, the probability that $i$ is in level $\ell + 1$ is at least $\frac{1}{2}$. $\qquad\square$

We next upper bound the expected search time for any item at any fixed level, where the randomness is over the construction of the skip list.

**Lemma B.2.** *In expectation, the search time for item $i \in [n]$ at level $\ell$ is at most $2$.*

*Proof.* Suppose item $i \in [n]$ is in level $\ell$. Let $S_{<i}^\ell \subseteq [n]$ be the subset of items in level $\ell$ that are less than $i$. Note that by Lemma B.1, each item of $S_{\leq i}^\ell$ is promoted to level $\ell + 1$ with probability at least $\frac{1}{2}$. Thus, the search time for item $i$ at level $\ell$ is $t$ if and only if the previous $t$ items in $S_{\leq i}^\ell$ were all not promoted, which can only happen with probability at most $\frac{1}{2^t}$. Hence, the expected search time $T$ for item $i \in [n]$ at level $\ell$ is at most

$$\mathbb{E}\left[T\right] \leq 1 \cdot \frac{1}{2} + 2 \cdot \frac{1}{2^2} + \ldots + n \cdot \frac{1}{2^n} \leq \sum_{t=1}^{\infty} \frac{t}{2^t} \leq 2.$$

$\qquad\square$

We now show that each item $i$ must be contained at some level depending on the predicted frequency $p_i$ of the item.

**Lemma B.3.** *Each item $i$ is included in level $\max\left(0, 1 + \lfloor \log(np_i) \rfloor\right)$.*

*Proof.* First, observe that all items are inserted at level $0$. Next, note that Algorithm 1 inserts item $i$ into level $\ell$ if $p_i \geq \frac{2^{\ell-1}}{n}$ or equivalently $\log(np_i) \geq \ell - 1$. Thus, each item $i$ is included in level $\max\left(0, 1 + \lfloor \log(np_i) \rfloor\right)$. $\qquad\square$

We next analyze the expected search time for each item $i$.

**Lemma B.4.** *Suppose the total number of levels is at most $C + \log n$ for some constant $C > 0$. Then the expected search time for item $i$ is at most $2C + 2\min\left(\log \frac{1}{p_i}, \log n\right)$.*

*Proof.* By Lemma B.3, item $i$ is included in level $\max\left(0, 1 + \lfloor \log(np_i) \rfloor\right)$. By Lemma B.2 the expected search time at each level is at most 2. Thus, in expectation, the total search time is at most $2(C + \log n - \max\left(0, 1 + \lfloor \log(np_i) \rfloor\right)) \le 2C + 2\min\left(\log\frac{1}{p_i}, \log n\right)$. $\qquad\square$

Finally, we analyze the expected search time across the true probability distribution $f_i$.

**Lemma B.5.** *Suppose the total number of levels is at most $C + \log n$ for some constant $C > 0$. For each $i \in [n]$, let $f_i$ be the proportion of queries to item $i$. Then the expected search time at most $2C + 2\sum_{i=1}^{n} f_i \cdot \min\left(\log\frac{1}{p_i}, \log n\right)$.*

*Proof.* For each query, the probability that the query is item $i$ is $f_i$. Conditioned on the total number of levels being at most $C + \log n$, then by Lemma B.4, the expected search time for item $i$ is at most $2C + 2\max\left(\log\frac{1}{p_i}, \log n\right)$. Thus, the expected search time at most

$$2C(f_1 + \ldots + f_n) + 2f_1 \min\left(\log\frac{1}{p_1}, \log n\right) + \ldots + 2f_n \min\left(\log\frac{1}{p_n}, \log n\right)$$

$$= 2C + 2\sum_{i=1}^{n} f_i \min\left(\log\frac{1}{p_i}, \log n\right).$$

$\square$

We now show that with high probability, the total number of levels in the skip list is at most $\mathcal{O}\left(\log n\right)$.

**Lemma B.6.** *With probability at least $0.99$, the total number of levels in the skip list is at most $10 + \log n$.*

*Proof.* For each level $\ell$, let $n_\ell$ be the number of items $i \in [n]$ that are deterministically promoted to exactly level $\ell$, i.e., $p_i \in \left[\frac{2^{\ell-1}}{n}, \frac{2^\ell}{n}\right)$. Note that for each fixed $i \in [n]$, the highest level it remains is a geometric random variable with parameter $\frac{1}{2}$, beyond the highest level at which it is deterministically placed. This is because the item is promoted to each higher level with probability $\frac{1}{2}$. Hence with probability $1 - \frac{1}{2^k}$, $i$ is not placed at least $k$ levels above its highest deterministic placement. Therefore, the probability that an item at level $\ell$ is placed at level $10 + \log n$ is at most $\frac{2^\ell}{1024n}$. Since no fixed $i$ will have predicted frequency more than 1, then no item will be deterministically placed at level $2 + \log n$. Hence by a union bound over all $\ell \in [2 + \log n]$, the probability that an item is placed at level $10 + \log n$ is at most

$$\sum_{\ell=0}^{2+\log n} \frac{n_\ell \cdot 2^\ell}{1024n}.$$

On the other hand, we have $\sum_{i=1}^{n} p_i = 1$, so that

$$\sum_{\ell=0}^{2+\log n} n_\ell \cdot 2^\ell \le 2n.$$

Therefore, with probability at least $0.99$, the total number of levels in the skip list is at most $10 + \log n$. $\qquad\square$

Thus, putting together Lemma B.5 and Lemma B.6, we get:

**Theorem 2.1.** *For each $i \in [n]$, let $f_i$ and $p_i$ be the proportion of true and predicted queries to item $i$. Then with probability at least $0.99$ over the randomness of the construction of the skip list, the expected search time over the choice of queries at most $20 + 2\sum_{i=1}^{n} f_i \cdot \min\left(\log\frac{1}{p_i}, \log n\right)$.*

## B.2 Near-Optimality

We first recall the construction of a Huffman code, a type of variable-length code that is often used for data compression. The encoding for a Huffman is known to be an optimal prefix code and can be represented by a binary tree, which we call the Huffman tree (Huffman, 1952).

To construct a Huffman code, we first create a min-heap priority queue that initially contains all the leaf nodes sorted by their frequencies, so that the least frequent items have the highest priority. The algorithm then iteratively removes the two nodes with the lowest frequencies from the priority queue, which become the left and right children of a new internal node that is created to represent the sum of the frequencies of the two nodes. This internal node is then added back to the priority queue. This process is continued until there only remains a single node left in the priority queue, which is then the root of the Huffman tree.

A binary code is then assigned to the paths from the root to each leaf node in the Huffman tree, so that each movement along a left edge in the tree corresponds to appending a 0 to the codeword, and each movement along a right edge in the tree corresponds to appending a 1 to the codeword. Thus, the resulting binary code for each item is the path from the root to the leaf node corresponding to the item.

Huffman coding is a type of symbol-by-symbol coding, where each individual item is separately encoded, as opposed to alternatives such as run-length encoding. It is known that Huffman coding is optimal among symbol-by-symbol coding with a known input probability distribution (Huffman, 1952) and moreover, by Shannon's source coding theorem, that the entropy of the probability distribution is an upper bound on the expected length of a codeword of a symbol-by-symbol coding:

**Theorem B.7** (Shannon's source coding theorem). *(Shannon, 2001) Given a random variable $X \in [n]$ so that $X = i$ with probability $f_i$, let $L(X)$ denote the length of the codeword assigned to $X$ by a Huffman code. Then $\mathbb{E}[L(x)] \geq H(f)$, where $H(f)$ is the entropy of $f$.*

We now prove our lower bound on the expected search time of an item drawn from a probability distribution $f$.

**Theorem 2.2.** *Given a random variable $X \in [n]$ so that $X = i$ with probability $f_i$, let $T(X)$ denote the search time for $X$ in a skip list. Then $\mathbb{E}[T(x)] \geq H(f)$, where $H(f)$ is the entropy of $f$.*

*Proof.* Let $\mathcal{L}$ be a skip list. We build a symbol-by-symbol encoding using the search process in $\mathcal{L}$. We begin at the top level. At each step, we either terminate, move to the next item at the current level, or move down to a lower level. Similar to the Huffman coding, we append a 0 to the codeword when we move down to a lower level, and we append a 1 to the codeword when we move to the next item at the current level. Now, the search time for an item $x$ in $\mathcal{L}$ corresponds to the length of the codeword of $x$ in the symbol-by-symbol encoding. By Theorem B.7 and the optimality of Huffman codes among symbol-by-symbol encodings, we have that $\mathbb{E}[T(x)] \geq H(f)$, where $f$ is the probability distribution vector of $x$. □

## B.3 Zipfian Distribution

In this section, we briefly describe the implications of our data structure to Zipfian distributions.

We first recall the following entropy upper bound for a probability distribution with support at most $n$.

**Theorem B.8.** *(Cover, 1999) Let $f$ be a probability distribution on a support of size $[n]$. Then $H(f) \leq \log n$.*

We can then upper bound the entropy of a probability vector that satisfies a Zipfian distribution with parameter $s$.

**Lemma 2.3.** *Let $s, z > 0$ be fixed constants and let $f$ be a frequency vector such that $f_i = \frac{z}{i^s}$ for all $i \in [n]$. If $s > 1$, then $H(f) = \mathcal{O}(1)$ and otherwise if $s \leq 1$, then $H(f) \leq \log n$.*

*Proof.* Since $f$ is a probability distribution on the support of size $[n]$, then by Theorem B.8, we have that $H(f) \leq \log n$. Thus, it remains to consider the case where $s > 1$. Since $z \leq 1$, we have

$$h(f) = \sum_{i=1}^{n} \frac{z}{i^s} \log \frac{i^s}{z}$$

$$\leq s \sum_{i=1}^{n} \frac{\log i}{i^s}.$$

Note that there exists an integer $\gamma > 0$ such that for $i > \gamma$, we have $\frac{\log i}{i^s} < \frac{1}{i^{(s+1)/2}}$. Since $s > 1$, then $\frac{s+1}{2} > 1$ and thus

$$\sum_{i=\gamma}^{n} \frac{1}{i^{(s+1)/2}} \leq \sum_{i=1}^{\infty} \frac{1}{i^{(s+1)/2}} = \mathcal{O}(1).$$

Hence,

$$h(f) \leq s \sum_{i=1}^{\gamma-1} \frac{\log i}{i^s} + s \sum_{\gamma}^{\infty} \frac{\log i}{i^s} = \mathcal{O}(1).$$

$\square$

By Theorem 2.1 and Lemma 2.3, we have the following statement about the performance of our learning-augmented skip list on a set of search queries that follows a Zipfian distribution.

**Corollary 2.4.** *With high probability, the expected search time on a set of queries that follows a Zipfian distribution with exponent $s$ is at most $\mathcal{O}(1)$ for $s > 1$ and $\mathcal{O}(\log n)$ for $s \leq 1$.*

### B.4 NOISY ROBUSTNESS

In this section, we show that our learning-augmented skip list construction is robust to somewhat inaccurate oracles. Let $f$ be the true-scaled frequency vector so that for each $i \in [n]$, $f_i$ is the probability that a random query corresponds to $i$. Let $p$ be the predicted frequency vector, so that for each $i \in [n]$, $p_i$ is the predicted probability that a random query corresponds to $i$. For $\alpha, \beta \in (0, 1)$, we call an oracle $(\alpha, \beta)$-noisy if for all $i \in [n]$, we have $p_i \geq \alpha \cdot f_i - \beta$.

**Lemma 2.5.** *Let $\alpha$ be a constant and $\beta < \frac{\alpha}{4n}$. A learning-augmented skip list with a set of $(\alpha, \beta)$-noisy predictions has performance that matches that of a learning-augmented learned with a perfect oracle, up to an additive constant.*

*Proof.* Suppose the total number of levels is at most $C + \log n$ for some constant $C > 0$. Note that this occurs with a high probability for a learning-augmented skip list with a set of $(\alpha, \beta)$-noisy predictions. For each $i \in [n]$, let $f_i$ be the proportion of queries to item $i$ and let $p_i$ be the predicted proportion of queries to item $i$. By Lemma B.5, the expected search time at most

$$2C + 2 \sum_{i=1}^{n} f_i \cdot \min\left(\log \frac{1}{p_i}, \log n\right).$$

Since the oracle is $(\alpha, \beta)$-noisy then we have $p_i \geq \alpha \cdot f_i - \beta$ for all $i \in [n]$.

We first note that in the expected search time for $i$ is proportional to $\min\left(\log \frac{1}{f_i}, \log n\right)$. Thus, for expected search time for item $i$, it suffices to assume $f_i > \frac{1}{2n}$ for all $i$.

Observe that for $f_i > \frac{1}{2n}$ and $\beta < \frac{\alpha}{4n}$, then $p_i \geq \alpha \cdot f_i - \beta$ implies

$$p_i \geq \alpha \cdot f_i - \beta \geq \alpha \cdot f_i - \frac{\alpha}{4n} \geq \frac{\alpha}{2} \cdot f_i.$$

Hence, we have $\frac{1}{p_i} \leq \frac{2}{\alpha} \cdot \frac{1}{f_i}$ so that the expected search time for item $i$ is at most

$$2C + 2 \cdot \min\left(\log \frac{1}{f_i} + \log \frac{2}{\alpha}, \log n\right).$$

Therefore, the expected search time is at most

$$2C + 2\sum_{i=1}^{n} f_i \cdot \min\left(\log \frac{1}{p_i}, \log n\right) \leq 2C + 2\sum_{i=1}^{n}\left(f_i \cdot \min\left(\log \frac{1}{f_i}, \log n\right) + f_i \cdot \log \frac{2}{\alpha}\right)$$

$$\leq 2C + 2\log \frac{2}{\alpha} + 2\sum_{i=1}^{n} f_i \cdot \min\left(\log \frac{1}{f_i}, \log n\right).$$

Since the perfect oracle would achieve runtime $2C + 2\sum_{i=1}^{n} f_i \cdot \min\left(\log \frac{1}{f_i}, \log n\right)$, then it follows that a learning-augmented skip list with a set of $(\alpha, \beta)$-noisy predictions has performance that matches that of a learning-augmented learned with a perfect oracle, up to an additive constant. $\square$

## C  MISSING PROOFS FROM SECTION 3

First, we will show that the expected depth of a given query depends on the probability of that query, and that high frequency queries must be found close to the root of our tree.

**Lemma C.1.** *Suppose $[\Delta]^d$ is the space of possible input points and queries. Let $N = \Delta^d$ and $p_i$ be the probability that a random query is made to $i \in [N]$, given the natural mapping between $[N]$ and $[\Delta]^d$. Then the level at which $i$ resides in the tree is at most $\mathcal{O}\left(\log \frac{1}{p_i}\right)$.*

*Proof.* First, consider only the high-frequency query points and data points for which we base our construction off.

In constructing the learning-augmented KD tree, we balance the contents of the children nodes such that a query to that node has a probability of $\frac{1}{2}$ of belonging to each of the children. Therefore, at a depth of $d$, the probability of belonging to either child is $\frac{1}{2^d}$. In particular, a query $i$ with probability $p_i$ at a depth $d$ must satisfy $p_i > \frac{1}{2^d}$. Thus, we have that the depth of $i$ is $\mathcal{O}\left(\log \frac{1}{p_i}\right)$, as desired.

Since the lowest probability of a high-frequency data point is $\frac{1}{n^2}$, this tree must have a depth of at most $2\log n$.

Now, consider a low-frequency data point, which we add to the bottom of the tree. By construction, the learned point of our tree has depth at most $2\log n$. Then, when inserting the additional data points as a balanced KD tree, we can accumulate at most an additional depth of $\log n$. Note, $p < \frac{1}{n^2}$ implies $\log n < \log \frac{1}{p}$. Thus, this low-frequency data point will have a depth of at most $3\log n = \mathcal{O}\left(\log \frac{1}{p}\right)$, as desired.

Similarly, if $i$ is not a data point and is low frequency, we achieve the same bound of $\mathcal{O}\left(\log \frac{1}{p}\right)$. In this case, we simply terminate at a leaf node and determine that the desired query is not in the dataset.

In summary, any query which has high frequency can be found in $\mathcal{O}\left(\log \frac{1}{p}\right)$ time. Low-frequency data points can similarly be found in $\mathcal{O}\left(\log \frac{1}{p}\right)$ time, and low frequency queries can be determined to not exist in $\mathcal{O}\left(\log \frac{1}{p}\right)$ time. $\square$

**Lemma C.2.** *Suppose $[\Delta]^d$ is the space of possible input points and queries. Let $N = \Delta^d$ and $p_i$ be the probability that a random query is made to $i \in [N]$, given the natural mapping between $[N]$ and $[\Delta]^d$. Then the level at which $i$ resides in the tree is at most $\mathcal{O}(\log n)$.*

*Proof.* This follows directly from the analysis in Llemma C.1. $\square$

Now, we have demonstrated the the depth of a given query point $i$ is bounded by both $\mathcal{O}(\log n)$ and $\mathcal{O}\left(\log \frac{1}{p_i}\right)$. Using this fact, we will now show that the expected query time of our algorithm is bounded by both the entropy of the dataset $H(p)$ in addition to $\log n$.

We now analyze the performance of our learning-augmented KD tree.

**Theorem 3.1.** *Suppose $[\Delta]^d$ is the space of possible input points and queries. Let $N = \Delta^d$ and $p_i$ be the probability that a random query is made to $i \in [N]$, given the natural mapping between $[N]$ and $[\Delta]^d$. Let $p = (p_1, \ldots, p_N) \in \mathbb{R}^N$ be the probability vector and $H(p)$ be its entropy. Then given a set of $n$ input points, the expected query time for the tree $T$ created by Algorithm 3 is $\mathcal{O}\left(\min(H(p), \log n)\right)$.*

*Proof.* Following C.1, the points $i$ in $[\Delta]^d$ with non-negligible probability $p_i \geq \frac{1}{n^2}$ are guaranteed to exist in the learning-augmented KD tree $\mathcal{T}$ with depth at most $\log \frac{1}{p_i}$. For points in the dataset $[n]$ with negligible probability, they exist in the tree and have depth in $\mathcal{O}\left(\log n\right)$. For all other points not contained in the KD tree, the query will terminate at a depth of $\mathcal{O}\left(\log n\right)$.

For any point $i$ in $[\Delta]^d$, the depth that a query to $i$ will terminate in the tree is

$$\text{Depth}(i) = \mathcal{O}\left(\min\left(\log \frac{1}{p_i}, \log n\right)\right). \tag{1}$$

Then, the expected search time $T$ is the expected depth of a given point,

$$\mathbb{E}\left[T\right] = \sum_{i \in [\Delta]^d} p_i \cdot \text{Depth}(i) = \sum_{i \in [\Delta]^d} p_i \cdot \mathcal{O}\left(\min\left(\log \frac{1}{p_i}, \log n\right)\right) = \mathcal{O}\left(\min(H(p), \log n)\right). \tag{2}$$

$\square$

Thus, we have shown that the expected query time of our algorithm is bounded by the entropy of the dataset. In particular, when the dataset has a highly skew distribution, $H(p)$ can be far less than $\log n$.

## C.1 NEAR-OPTIMALITY

Near-optimality of our learning-augmented KD trees uses a similar argument to the proof of the near-optimality of our learning-augmneted skip lists. In particular, we again utilize Shannon's source coding theorem from Theorem B.7. We then have the following:

**Theorem 3.2.** *Given a random variable $X \in [n]$ so that $X = i$ with probability $f_i$, let $D(X)$ denote the depth for $X$ in a learning-augmented KD tree. Then $\mathbb{E}\left[D(X)\right] \geq H(f)$, where $H(f)$ is the entropy of $f$.*

*Proof.* In a learning-augmented KD tree, the search path to an element $i$ can be encoded as a $0 - 1$ codeword, with entries indicating whether the lower or upper branch is taken at each node traversal. Moreover, the length of this codeword in the symbol-by-symbol encoding corresponds to the depth of element $i$. Then, by Theorem B.7 and the optimality of Huffman codes in symbol-by-symbol encodings, we have that $\mathbb{E}\left[D(X)\right] \geq H(f)$, as desired. $\square$

## C.2 NOISY ROBUSTNESS

In this section, we analyze the performance of the learning-augmented KD tree under noisy data conditions.

Previously, we analyzed the performance of the learning-augmented KD-tree assuming access to a perfect prediction oracle.

Now we analyze the performance with a noisy oracle. That is, for each $i \in [\Delta]^d$ there is a true-scaled frequency $f_i$ that the point will be queried and that $p_i$ is a prediction made by the noisy oracle.

First, we analyze the multiplicative robustness of the algorithm. In this case, the oracle predicts $f_i$ up to some multiplicative constant $\alpha \in \mathbb{R}^+$ such that $f_i = \alpha \, p_i$.

**Lemma C.3.** *Suppose $[\Delta]^d$ is the space of possible input points and queries. Let $N = \Delta^d$ and $p_i$ be the probability that a random query is made to $i \in [N]$ during tree construction. Suppose during runtime that the true probability of querying $i$ is $f_i = \alpha \, p_i$ for some $\alpha \in \mathbb{R}^+$. Then the level at which $i$ resides in the tree is at most $\mathcal{O}\left(\log \frac{1}{f_i} + \log \frac{1}{\alpha}\right)$.*

*Proof.* If $\alpha \leq 1$, this is immediate. If this is the case, in construction we expected $i$ to be queried more often than it actually is, so our construction placed the point $i$ higher in the tree than is necessary. Thus, the depth is at most the previously shown $\frac{1}{p_i} \leq \frac{1}{f_i}$.

Now, suppose $\alpha > 1$. In this case, we must have placed $i$ deeper in the tree than we should have, as our construction frequency is less than the true query frequency. Then, as in C.1, the depth of $i$ is $\mathcal{O}\left(\log \frac{1}{p_i}\right)$. Now, we have that

$$\mathcal{O}\left(\log \frac{1}{p_i}\right) = \mathcal{O}\left(\log \frac{\alpha}{f_i}\right) = \mathcal{O}\left(\log \frac{1}{f_i} + \log \alpha\right). \tag{3}$$

$\square$

Having shown multiplicative robustness of the learning-augmented KD tree we next analyze the additive-multiplicative robustness of the method. An oracle is $(\alpha, \beta)$-noisy if the prediction satisfies $p_i \geq \alpha f_i - \beta$ for constants $\alpha, \beta \in (0, 1)$.

**Lemma 3.4.** *Let $\alpha$ be a constant and let $\beta \leq \frac{\alpha}{n^2}$. Then the query time for our learning-augmented KD tree with $(\alpha, \beta)$-noisy prediction matches the performance of a learning-augmented KD tree constructed using a perfect oracle up to an additive constant.*

*Proof.* From Lemma C.1 we have that the depth of a point $i$ in the learning-augmented KD tree is $\mathcal{O}\left(\frac{1}{p_i}\right)$ where $p_i$ is the predicted frequency of $i$.

Then, with an $(\alpha, \beta)$-noisy oracle we have that the prediction is bounded from below by $p_i \geq \alpha f_i - \beta$. As $i$ is a point in the tree, we can assume that the true-scaled frequency $f_i$ has a lower-bound of $1/n^2$.

By choosing $\beta \leq \alpha/n^2$ we ensure that the predicted $p_i$ is always nonnegative. Then, let $\beta = \alpha/2n^2$

$$p_i \geq \alpha f_i - \beta \geq \alpha f_i - \frac{\alpha}{2n^2} \geq \frac{\alpha}{2} f_i \tag{4}$$

Then, applying Lemma C.1 again

$$\mathcal{O}\left(\log \frac{1}{p_i}\right) = \mathcal{O}\left(\log \frac{2}{\alpha f_i}\right) = \mathcal{O}\left(\log \frac{1}{f_i} + \log \frac{1}{\alpha}\right). \tag{5}$$

$\square$

# D ADDITIONAL EMPIRICAL EVALUATIONS

## D.1 SKIP LISTS

In this section, we perform empirical evaluations comparing the performance of our learning-augmented skip list to that of traditional skip lists, on both synthetic and real-world datasets. Firstly, we compare the performance of traditional skip lists with our learning-augmented skip lists on synthetically generated data following Zipfian distributions. The proposed learning-augmented skip lists are evaluated empirically with both synthetic datasets and real-world internet flow datasets from the Center for Applied Internet Data Analysis (CAIDA) and AOL. In the synthetic datasets, a diverse range of element distributions, which are characterized by the skewness of the datasets, are evaluated to assess the effectiveness of the learning augmentation. In the CAIDA datasets, the $\alpha$ factor is calculated to reflect the skewness of the data distribution.

The metrics of performance evaluations include insertion time and query time, representing the total time it takes to insert all elements in the query stream and the time it takes to find all elements in the query stream using the data structure, respectively.

The computer used for benchmarking is a Lenovo Thinkpad P15 with an intel core i7-11800H@2.3GHz, 64GB RAM, and 1TB of Solid State Drive. The tests were conducted in a Ubuntu 22.04.3 LTS OS. GNOME version 42.9.

### D.1.1 SYNTHETIC DATASETS

In the synthetic datasets, both the classic and augmented skip lists are tested against different element counts and $\alpha$ values. In terms of the distribution of the synthetic datasets, the uniform distribution and a Zipfian distribution of $\alpha$ between 1.01 and 2 with query counts up to 4 million are evaluated. It is worth noting that the number of unique element queries could vary for the same query count at different $\alpha$ values in the Zipfian distribution, which may affect the insertion time.

Table 1: Speed up factor of augmented skip list over classic skip list under different synthetic distributions

| Distribution | Query size of synthetic data (unit: thousand) | | | | | | | | | | | |
|---|---|---|---|---|---|---|---|---|---|---|---|---|
| | 0.5 | 10 | 100 | 500 | 1000 | 1500 | 2000 | 2500 | 3000 | 3500 | 4000 | Average |
| uniform | 3.02 | 0.84 | 1.01 | 1.05 | 1.11 | 1.14 | 1.17 | 1.21 | 1.22 | 1.42 | 1.4 | 1.33 |
| $\alpha$=1.01 | 3.63 | 2.6 | 1.04 | 1.24 | 1.03 | 1.21 | 1.2 | 1.14 | 1.3 | 1.18 | 1.3 | 1.53 |
| $\alpha$=1.25 | 3.28 | 3.74 | 5.87 | 2.89 | 2.47 | 3.21 | 2.95 | 3.34 | 3.55 | 3.16 | 3.12 | 3.42 |
| $\alpha$=1.5 | 2.42 | 8.97 | 6.93 | 6.54 | 7.99 | 5.83 | 4.65 | 3.8 | 4.92 | 5.34 | 5.93 | 5.76 |
| $\alpha$=1.75 | 12.43 | 10.4 | 5.76 | 9.78 | 6.76 | 7.13 | 7.31 | 7.09 | 6.63 | 5.07 | 6.98 | 7.76 |
| $\alpha$=2 | 8.19 | 2.5 | 5.56 | 10.1 | 4.47 | 3.91 | 7.26 | 5.33 | 9.29 | 7.65 | 5.55 | 6.35 |

Table 2: Node count for each distribution configuration in the 4 million dataset

| $\alpha$ | Unique node count |
|---|---|
| 1.01 | 2886467 |
| 1.25 | 259892 |
| 1.75 | 8386 |
| 2 | 2796 |

Table 1 shows the speed-up factor, defined as the time taken by the augmented skip list over the classic skip list for the same query stream. We can observe a progressive improvement in the performance of our augmented skip lists as the dataset skewness increases. It also suggests that our augmented skip list will perform at least as good as the traditional skip list and will outperform a traditional skip list by a factor of up to 7 times depending on the skewness of the datasets.

Figure 6 shows that the insertion time decreases with more skewed datasets for the same size of the query stream. This is attributed to the reduced number of nodes in the datasets, as shown in Table 2.

The query time of augmented skip lists is also reduced greatly compared to the classic skip lists as shown in Figure 7.

In addition, we conduct experiments to compare the performance of standard binary search trees and standard skip lists. In particular, we generate datasets of size $n \in \{5000, 10000, 15000, 20000, 25000, 30000, 35000, 40000, 45000, 50000, \}$. For each fixed value of $n$, each element of the dataset is generated uniformly at random in $[2n]$, i.e., uniformly at random from $\{1, 2, \ldots, 2n\}$. Because the dataset is generated uniformly at random, then learning-augmented data structures will perform similar to oblivious data structures. We measure the construction time of the data structures, based on the input dataset. Our results demonstrate that as expected, skip lists perform significantly better than balanced binary search trees across all values of $n$, due to the latter's necessity of constantly rebalancing the data structure. In fact, skip lists performed almost $4\times$ better than BSTs in some cases, e.g., $n = 20000$. We illustrate our results in Figure 8.

### D.1.2 AOL DATASET

The AOL dataset (G. Pass, 2006) features around 20M web queries collected from 650k users over three months. The distribution of the queries is shown in Figure 9. The AOL dataset is a less skewed dataset than CAIDA with an alpha value of 0.75.

Figure 9 shows the distribution of the AOL queries with an estimated alpha value of 0.75. The AOL dataset resembles more to a slightly skewed uniform distribution with very few highly frequent items, which accounts for a lower improvement as in the case of AOL shown in Figure 10. The total number of queries for items with higher than 1000 frequency accounts for only 5% of the total number of queries for the AOL datasets. The learning-augmented skip list still outperforms the traditional skip

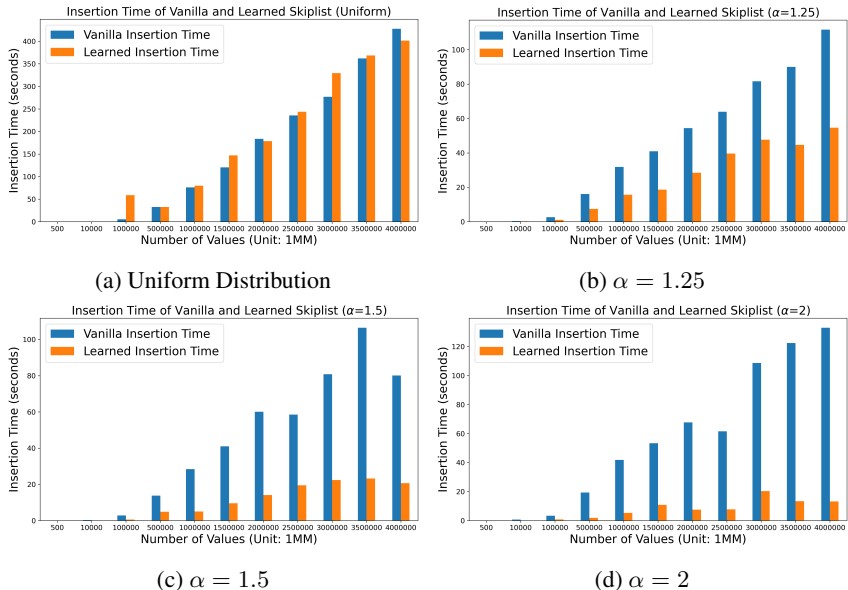

Figure 6: Insertion time for synthetic datasets with a uniform distribution and under different $\alpha$ values of the Zipfian distribution for both classic and augmented skip lists. This figure illustrates the insertion time on the synthetic data for both the uniform distribution and the Zipfian distribution at different $\alpha$ values. Generally, higher skewness of the datasets results in less insertion time when using the augmented structure. The decrease in insertion time is proportional to the increase in the $\alpha$ value, as a higher $\alpha$ value leads to a reduction in the number of unique nodes, as illustrated in Table 2.

list on this slightly skewed dataset. This result is also in line with the results from the synthetic data shown in Table 1 where lower alpha values have resulted in a lower speedup factor.

## D.2 KD TREES

For KD Trees, first describe our methodology for evaluating our data structure on synthetic data. We then describe our empirical evaluations on real-world datasets.

The computer used for KD tree benchmarking is a desktop machine with an Intel Core i9-14900KF@ 3200MHz, with 64GB RAM, and 2TB of Solid State Drive. The tests were conducted in Windows 10 Enterprise, version 10.0.19045 Build 19045.

### D.2.1 SYNTHETIC DATASETS WITH PERFECT KNOWLEDGE

First, consider a dataset with a Zipfian distribution. In order to construct this dataset, we first select $n$ unique data points in $[\Delta]^d$ uniformly. We then generate Zipfian frequencies, $f_i \approx \frac{1}{(i+b)^a}$, and randomly pair the frequencies to the data points to serve as both the construction and query frequencies. We construct either a traditional KD tree, or our learned KD tree on this dataset. Then, we evaluate the performance of querying by sampling the known datapoints with probabilities given by their Zipfian probabilities.

In Table 3, we use a Zipfian distribution with parameters $a = 1$ and $b = 2.7$. This data demonstrates that our method outperforms a traditional KD tree, given that the Zipfian distribution has the given parameters. In Figure 11, we vary the Zipfian parameters of our dataset. As expected, our learned KD tree performance increases with how skew the distribution is. Moreover, we find that our method outperforms the traditional KD tree on all tested Zipfian distributions.

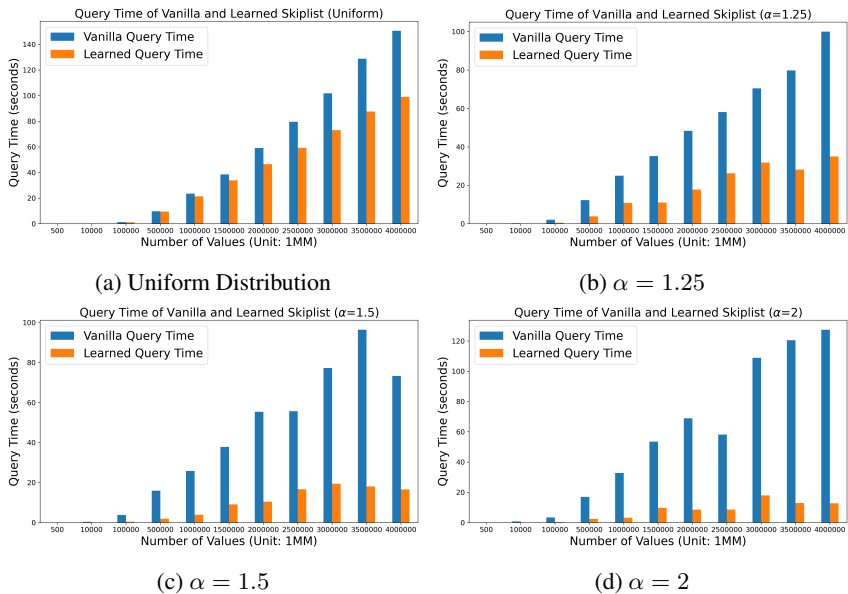

(a) Uniform Distribution

(b) $\alpha = 1.25$

(c) $\alpha = 1.5$

(d) $\alpha = 2$

Figure 7: Query time for synthetic datasets with a uniform distribution and under different $\alpha$ values of the Zipfian distribution for both classic and augmented skip lists. This figure compares the query time of the classic and augmented skip lists for both uniform distribution and Zipfian distribution at various $\alpha$ values. Similar to insertion, the query time is significantly reduced under different query sizes with the implemented augmentation. The performance enhancement is especially pronounced for the high skewness of the dataset.

| type | dim | const_time | avg query(s) | avg query depth |
|---|---|---|---|---|
| traditional | 1 | 1.267E-01 | 2.645E-05 | 1.330E+01 |
| learned | 1 | 2.387E-01 | 2.181E-05 | 1.086E+01 |
| traditional | 2 | 1.266E-01 | 2.709E-05 | 1.341E+01 |
| learned | 2 | 3.232E-01 | 2.221E-05 | 1.086E+01 |
| traditional | 3 | 1.253E-01 | 2.700E-05 | 1.334E+01 |
| learned | 3 | 3.981E-01 | 2.264E-05 | 1.097E+01 |
| traditional | 4 | 1.185E-01 | 2.724E-05 | 1.336E+01 |
| learned | 4 | 4.549E-01 | 2.256E-05 | 1.089E+01 |
| traditional | 5 | 1.300E-01 | 2.743E-05 | 1.336E+01 |
| learned | 5 | 5.286E-01 | 2.296E-05 | 1.096E+01 |
| traditional | 10 | 1.277E-01 | 2.896E-05 | 1.340E+01 |
| learned | 10 | 8.642E-01 | 2.403E-05 | 1.091E+01 |
| traditional | 20 | 1.295E-01 | 3.121E-05 | 1.344E+01 |
| learned | 20 | 1.543E+00 | 2.564E-05 | 1.085E+01 |
| traditional | 40 | 1.361E-01 | 3.558E-05 | 1.340E+01 |
| learned | 40 | 2.911E+00 | 2.902E-05 | 1.078E+01 |

Table 3: We construct and query KD trees with our method and with a traditional KD tree on synthetic datasets of various dimensionality. We construct our tree on 10k points, and query 1M times. We find that, independent of the data dimensionality, our method produces lower average query depths.

### D.2.2 SYNTHETIC DATASETS WITH NOISY KNOWLEDGE

In reality, it is rarely the case that we have perfect knowledge in constructing a model. In order to evaluate the performance of our method on noisy data, we create a synthetic dataset with noisy training information.

As before, we generate world points and assign them Zipfian weights. When constructing the tree, each weight us updated to be $Mp_i + A$, where $M$ and $A$ are drawn from uniform distributions. This

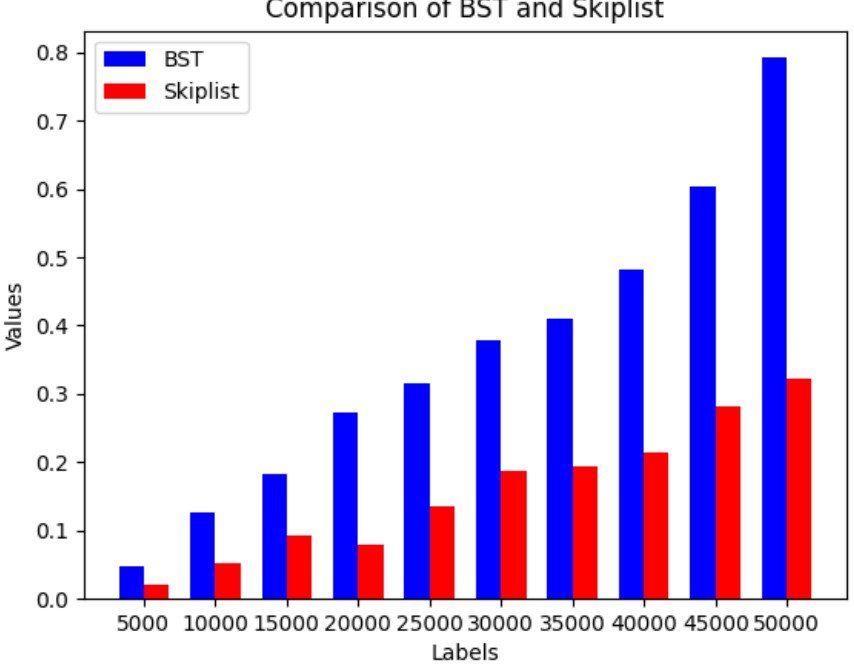

Figure 8: BST vs skip list construction times

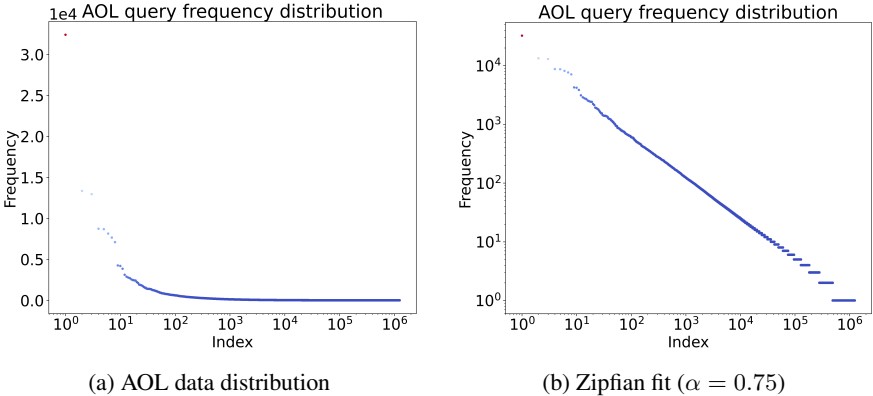

(a) AOL data distribution

(b) Zipfian fit ($\alpha = 0.75$)

Figure 9: AOL datasets distribution characterization. This figure illustrates how the $\alpha$ value of 0.75 is obtained for the AOL dataset. The AOL dataset shows a much smaller $\alpha$ value compared to the CAIDA dataset so AOL almost resembles a uniform distribution despite very few high-frequency nodes. This also explains why the performance of the augmented skip list is close to the classic implementation.

new noisy distribution is normalized in order to form a valid probability distribution. Then, points are queried many times with their ground truth Zipfian probabilities. For the fixed Zipfian distribution with $a = 5, b = 2$, we plot the effects of different ranges of $M$ and $A$ to demonstrate the effect noise has in Fig. 4. This figure demonstrates that, even with moderate amounts of noise, our method still outperforms a traditional KD tree. Moreover, our method still remains on par with the traditional KD tree when significant noise is present, due to our robustness guarantees.

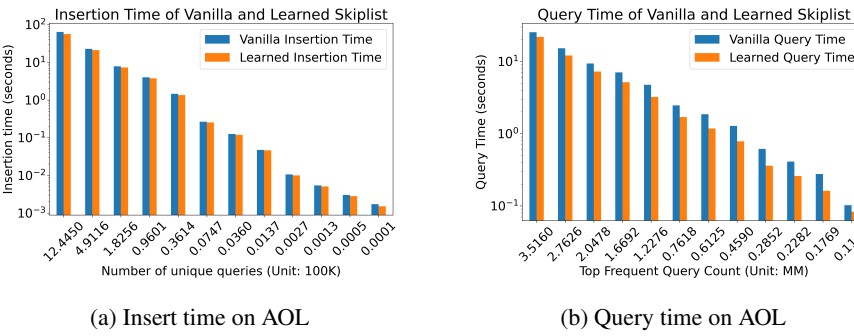

(a) Insert time on AOL                    (b) Query time on AOL

Figure 10: Insertion and query time on AOL of classic and augmented skip lists

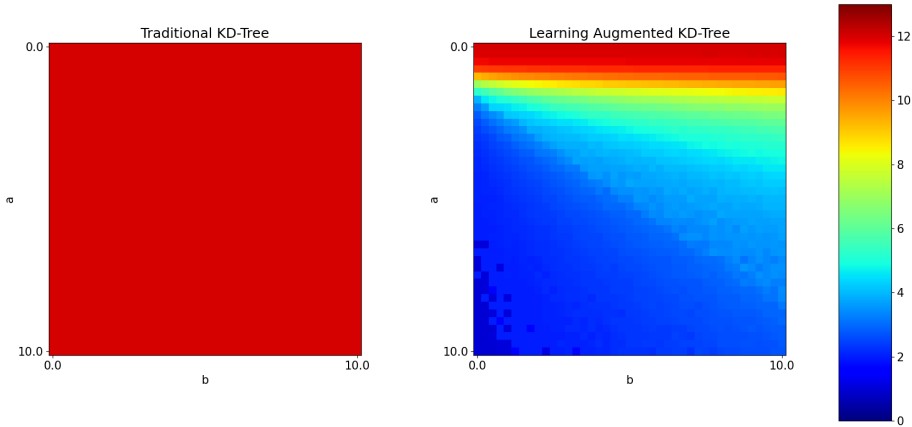

Figure 11: We generate datasets of $2^{12}$ points in 3-dimentional space, with frequencies given by a Zipfian distribution with parameters $a, b$. We then query the tree $2^{14}$ times, with point queries selected by the same Zipfian distribution. We repeat this process 32 times, and report the median of the average query depth across all runs. We find that our method is able to outperform the traditional KD tree method across all tested Zipfian distributions. Moreover, our performances increases as $a$ increases and $b$ decreases, making the Zipfian distribution most skew. Notably, when $a = 0$, all points have uniform weights, at which point our method performs equivalently to the traditional KD tree.

### D.2.3  REAL-WORLD DATASETS

In addition to evaluating our method on synthetic data, we also evaluate results on real-world data.

First, we consider $n$-grams in various languages. We test on a pre-processed subset of the Google N-Gram dataset (Goog; Google, 2012).

In order to evaluate our method, we convert an $n$-gram to a vector in $\mathbb{Z}^n$ with each entry indexing the words in the $n$-gram. We construct the learning-augmented and traditional KD trees, and show the performance of lookup with queries weighted by their ground truth frequency in Table 4. In all cases, we find that the learning-augmented KD tree outperforms the traditional KD tree at average lookup depth.

Additionally, we test our method on a dataset of neuron activity, as provided by Aitchison et al. (2014), which has shown to be Zipfian. This dataset consists of vectors in $\{0, 1\}^{30}$, indicating which of 30 cells fire at agiven time. As in their work, we bin in 20ms increments when constructing these

vectors. We similarly ignore the time of observations, and build our learning-augmented KD tree with frequencies given by the rate of appearance of vectors. We find that, when querying with probabilities equivalent to the training distribution, a traditional KD tree has an average query depth of 23.7. Using our learning-augmented KD tree, however, we are able to achieve an average query depth of 14.9.

| Dataset | Traditional Avg Query Depth | Learned Avg Query Depth |
|---|---|---|
| 2grams_chinese_simplified | 14.8388 | 12.1144 |
| 2grams_english-fiction | 14.216 | 11.6232 |
| 2grams_english | 14.7253 | 11.6647 |
| 2grams_french | 14.5123 | 11.5999 |
| 2grams_german | 13.9473 | 11.7066 |
| 2grams_hebrew | 13.5322 | 12.0007 |
| 2grams_italian | 14.5231 | 11.8027 |
| 2grams_russian | 14.5769 | 11.8082 |
| 2grams_spanish | 15.1763 | 11.6582 |
| 3grams_chinese_simplified | 12.343 | 11.4211 |
| 3grams_english-fiction | 13.5264 | 11.3183 |
| 3grams_english | 15.002 | 11.3632 |
| 3grams_french | 14.9235 | 11.3529 |
| 3grams_german | 14.129 | 11.4285 |
| 3grams_hebrew | 13.0887 | 9.6663 |
| 3grams_italian | 13.3839 | 11.3616 |
| 3grams_russian | 15.2475 | 11.1415 |
| 3grams_spanish | 15.3855 | 11.3635 |
| 4grams_chinese_simplified | 11.5823 | 9.8661 |
| 4grams_english-fiction | 12.5141 | 9.7589 |
| 4grams_english | 13.4464 | 9.7793 |
| 4grams_french | 12.6383 | 9.835 |
| 4grams_german | 12.3911 | 9.8765 |
| 4grams_hebrew | 9.3198 | 7.4493 |
| 4grams_italian | 11.3456 | 9.6834 |
| 4grams_russian | 13.2406 | 9.6274 |
| 4grams_spanish | 12.9712 | 9.8191 |
| 5grams_chinese_simplified | 11.5373 | 9.8982 |
| 5grams_english-fiction | 12.8027 | 9.8069 |
| 5grams_english | 12.5607 | 9.5967 |
| 5grams_french | 12.402 | 9.835 |
| 5grams_german | 11.6912 | 9.8432 |
| 5grams_hebrew | 7.097 | 6.2143 |
| 5grams_italian | 11.3061 | 9.7771 |
| 5grams_russian | 12.885 | 9.6805 |
| 5grams_spanish | 12.111 | 9.8079 |

Table 4: We construct traditional and learning-augmented KD trees for $n$-grams for various languages, and of various lengths $n$. Our method outperforms a traditional KD tree in all cases.

