# OpenReview forum: "Learning-Augmented Search Data Structures"
_ICLR.cc/2025/Conference — ICLR 2025 Poster_

### Official Review · Reviewer_zTZd · 2024-11-02

**Soundness:** 3
**Presentation:** 3
**Contribution:** 2
**Rating:** 8
**Confidence:** 3

**Summary:**

The paper studies search data structures augmented with an "oracle" (some ML-based prediction) of query frequencies, which can be used to improve over the classical setting that assumes no foresight into the query distribution. It focuses specifically on skip lists and KD-trees. It develops ML-augmented analogs, proves bounds on their performance under perfect and imperfect predictions, and presents an empirical evaluation.

**Strengths:**

The problems are interesting and relevant, the theoretical results seem solid, and the empirical results show improvement. It is overall a solid submission.

**Weaknesses:**

-- No related work section nor discussion, except for a couple of very specifically related recent/concurrent papers. No broader positioning in the literature; even though there is past work on very similar topics which goes unmentioned and unreferenced (like learning-augmented KD-trees in Cayton et al., see below).

-- The font size in the figures is beyond excusable

**Questions:**

How do your methods and results compare to the learning-based data structures of Ailon et al. Cayton et al., and other works related to those? They seem to study very similar problems in related settings.

Ailon, N., Chazelle, B., Clarkson, K. L., Liu, D., Mulzer, W., & Seshadhri, C. (2011). Self-improving algorithms. SIAM Journal on Computing, 40(2), 350-375.

Cayton, L., & Dasgupta, S. (2007). A learning framework for nearest neighbor search. Advances in Neural Information Processing Systems, 20.

---

> ### Author Response · Authors · 2024-11-20
>
> > No related work section nor discussion, except for a couple of very specifically related recent/concurrent papers. No broader positioning in the literature; even though there is past work on very similar topics which goes unmentioned and unreferenced (like learning-augmented KD-trees in Cayton et al., see below).
>
> Thanks for the comment. We have significantly expanded the discussion on related work, adding an additional section in the appendix in the updated version of the manuscript, mentioning works in learning-augmented algorithms, classic literature on data structures, and some more recent work, such as the references pointed out by the reviewer.
>
> > The font size in the figures is beyond excusable
>
> We apologize for the font sizes. We have recreated the images in the updated version of the manuscript, using larger font sizes for both the titles and the labels. Please let us know if they should be further increased.
>
> > How do your methods and results compare to the learning-based data structures of Ailon et al. Cayton et al., and other works related to those? They seem to study very similar problems in related settings.
>
> > Ailon, N., Chazelle, B., Clarkson, K. L., Liu, D., Mulzer, W., & Seshadhri, C. (2011). Self-improving algorithms. SIAM Journal on Computing, 40(2), 350-375.
>
> > Cayton, L., & Dasgupta, S. (2007). A learning framework for nearest neighbor search. Advances in Neural Information Processing Systems, 20.
>
> Whereas our paper studies how to utilize an oracle for learning-augmented data structures, the paper by Cayton et. al. (2007) studies how to obtain such an oracle. In particular, they study generalization bounds in the context of learning theory, analyzing the number of samples from an underlying distribution necessary to produce an oracle with a small error rate.
>
> On the other hand, Ailon, et. al. (2011) studies algorithms for sorting and clustering that can improve their expected performance given access to multiple instances sampled from a fixed distribution. Although the high-level goal of improving algorithmic performance using auxiliary information is the same as ours, the specifics of the paper seem quite different than ours, as the paper focuses on techniques for sorting and clustering.
>
> Thanks for pointing out these references. We have incorporated discussion of both of these works (among others) in the additional related works in the appendix of the updated version of the manuscript.

---

> > ### Comment · Reviewer_zTZd · 2024-11-21
> >
> > Thank you for the answers and the updates.

---

### Official Review · Reviewer_i7vt · 2024-11-06

**Soundness:** 3
**Presentation:** 4
**Contribution:** 2
**Rating:** 8
**Confidence:** 3

**Summary:**

The  studies using machine learning models for improving other machine learning algorithms, namely search algorithms that use efficient data structures to organize data elements.

The paper considers skip lists and KD-trees and augments them by adding an output of a learning algorithm. Skip lists organize data probabilistically in hierarchical and traversal is similar to binary search but with larger number of splits.

The paper consider unbalanced data distributions, mentioning that unbalanced distributions may result in suboptimal balancing behavior when underlying data structure construction algorithm expects balanced data. Authors propose to use an external machine learning algorithm that predicts statistics of the input data and provides an information to adjust the baseline search algorithm and data structure.

Proposed learning augmented skip list uses predicted probabilities to promote items to next levels with exponentially increasing required thresholds for predicted probabilities.

While it may seem reasonable to use simple frequency estimation when number of possible items is not large, when number of items in the dataset is larger than number of encountered queries, having a machine learning oracle estimate expected frequency of previously unseen queries is a good idea. The work allows using such oracles to improve search data structures and proves better bounds on search time in general case and in case of an important and popular in real world datasets Zipfian distribution; and proves similar bounds for noisy oracle (representing real machine learning model with positive probability of incorrect guesses).

The idea behind learning augmented KD-tree is to make sure that each node or leaf has same probability on each tree height. Oracle predicted probabilities are used to ensure balanced splitting, improving search time if the oracle is usually correct. To limit algorithm complexity, low probability items are inserted into the tree using conventional approach. Similar bounds, as in case of skip lists, are proven for the augmented KD-tree.

Empirically, the paper shows constant factor speedup of the augmented data structures, compared to the conventional approach

**Strengths:**

- Provided approach is an elegant way to improve search data structure performance by considering probabilistic nature of the data.
- Given the abundance of machine learning and statistical methods to serve as an oracle, proposed approach can be widely used in practice.
- New bounds were proven theoretically and experimental results partially support the theoretical findings.

**Weaknesses:**

- From the paper it was not fully clear how to use proposed method with real machine learning oracle models (that non-trivially predict frequencies), instead of statistical oracles (that just calculate the table of frequencies).
- Some graphs, such as Figure 2, show a constant factor speed up. It would be nice to clarify what is theoretically predicted speedup in the cases of various datasets and parameters, and how theory aligns with practice.

**Questions:**

- In the experiments, as I understood, simple history-based oracles were used. Is it the main use-case of the proposed method, or do we expect large oracles, such as deep learning models to be used as well?
- If the main use case will be using large machine learning oracles, I have the following question: for example, for image search, an oracle can be a neural network that takes an image and outputs expected query frequency; or for text search it could be a language model that outputs expected frequency of text or part of the text -- will the accuracy of real-world machine learning models be enough to guarantee improvement over conventional search structures?
- Would be nice to present graphs with theoretically expected improvement and actual improvement in various settings.

---

> ### Author Response · Authors · 2024-11-20
>
> > From the paper it was not fully clear how to use proposed method with real machine learning oracle models (that non-trivially predict frequencies), instead of statistical oracles (that just calculate the table of frequencies).
>
> Indeed, there are many possible different machine learning oracle models that could return various forms of prediction. For example, the oracle could predict the "depth" an item should be in a specific search data structure. The oracle could also output a wealth of information from which a frequency prediction can be extracted. Utlimately, the algorithms corresponding to complex machine learning oracle models can change depending on the way these models predict frequencies and how the predictions interact with the proposed method. We definitely agree that the real machine learning oracle models that data analysts find to be the most useful is an interesting research direction.
>
> > Some graphs, such as Figure 2, show a constant factor speed up. It would be nice to clarify what is theoretically predicted speedup in the cases of various datasets and parameters, and how theory aligns with practice.
>
> > Would be nice to present graphs with theoretically expected improvement and actual improvement in various settings.
>
> Thanks for the suggestion. We looked at implementing various versions of this. For example, the theoretical guarantees for our learning-augmneted skiplist currently states that with probability $0.99$ (actually the analysis shows probability $0.999$), the expected runtime of a query is at most $20+2\cdot H(\alpha)$. For the CAIDA dataset, which aligns closely with a Zipfian distribution with exponent roughly $\alpha\approx 1.37$, this is roughly $27.5$. The size of the dataset is roughly $n=30$ million, so the expected runtime of a query is at most $2\log n\approx 49.7$. The theoretical speed-up is therefore $1.81\times$. Indeed our actual speed-up in Figure 2b is $1.86\times$, which is surprisingly close.
>
> Despite the closeness of the theoretically predicted speedup versus the actual speedup in this case, the general graph may not be as insightful, due to the multiple parameters in the theoretical guarantees. For example, if we want a extremely probability of success, say $1-\frac{1}{10^{10}}$, then the expected runtime of a query in our learning-augmented skiplist would be quite large (in this case $66+2\cdot H(\alpha)$, which may be even larger than $2\log n$ for small datasets, i.e., datasets with small $n$. On the other hand, if we just want our guarantees to hold with probability $\frac{2}{3}$, then it can be shown that the expected runtime of a query in our learning-augmented skiplist is at most $4+2\cdot H(\alpha)$, which yields a different ratio between the theoretical and actual speed-up. Finally, our experiments in Figure 2b are wall-clock query time, which includes additional operations besides the number of nodes visited, so the ratios do not correspond exactly to the ratio between the theoretical and actual speed-up.
>
> For these reasons, we have added discussion of the theoretical and actual speed-up in the appendix of the updated version, but we have not added a graph or table, though we would be happy to do so, if the reviewer sees additional value in such a figure in light of these points.
>
> > In the experiments, as I understood, simple history-based oracles were used. Is it the main use-case of the proposed method, or do we expect large oracles, such as deep learning models to be used as well?
>
> In our experiments, our oracles have multiple sources of error. Firstly, we considered history-based oracles from a separate dataset, so that the predicted frequencies do not match the actual frequencies (and for some items could be quite different). Secondly, we add noise to the frequencies predicted by the oracle. Thus our oracle implementations are quite simple and noisy, and it is our hope that large-scale deep learning models will only do better than our simple oracles. More generally, our theoretical guarantees show that any oracle implementation that provides $(\alpha,\beta)$-noisy estimates for the predictions will provably achieve some degree of optimality given by the $(\alpha,\beta)$ parameters.

---

> > ### Author Response · Authors · 2024-11-20
> >
> > > If the main use case will be using large machine learning oracles, I have the following question: for example, for image search, an oracle can be a neural network that takes an image and outputs expected query frequency; or for text search it could be a language model that outputs expected frequency of text or part of the text -- will the accuracy of real-world machine learning models be enough to guarantee improvement over conventional search structures?
> >
> > The accuracy of machine learning models for simple tasks like predicting query frequency can, in many cases, surpass conventional search structures by capturing complex patterns and semantics. However, improvement is not guaranteed, as real-world performance depends on factors like data quality and model training. While machine learning oracles offer flexibility and adaptability, their benefits must be weighed against the efficiency and reliability of traditional methods, making the outcome highly context-dependent. Regardless, our theoretical guarantees demonstrate both consistency, i.e., performance beyond conventional search structures when real-world machine learning models exhibit high quality, and robustness, i.e., performance competitive with conventional search structures when real-world machine learning models perform poorly.

---

> > > ### Comment · Reviewer_i7vt · 2024-12-03
> > >
> > > Thank you for the updates and a detailed response, I am keeping my high score and recommendation.

---

### Official Review · Reviewer_Azhq · 2024-11-08

**Soundness:** 4
**Presentation:** 3
**Contribution:** 3
**Rating:** 6
**Confidence:** 4

**Summary:**

The paper proposes learning-augmented search data structures based on skip lists and KD trees. The authors achieve optimal expected search time and demonstrate robustness to inaccurate frequency predictors. Experimental results verify the effectiveness of the proposed algorithms.

**Strengths:**

1. The integration of learned frequencies into skip lists and KD trees is well-constructed and achieves optimal performance.
2. Experiments show that the proposed algorithm outperforms classical algorithms and is robust to noise.
3. The paper is clearly written and easy to read. The ideas are simple, yet novel and effective. I appreciate the clear comparison

**Weaknesses:**

1. The authors slightly overstate their contributions. For example, the claim of constant expected search time under the Zipfian distribution holds only when the exponent $s>1$, which is due to the entropy being of constant order (Lemma 2.3). In other words, for any data structure that achieves optimality has the same property. Additionally, in the abstract, the claim on the robustness when predictions are arbitrarily incorrect is not what you describe later.
2. The noise robustness measure, denoted as $(\alpha,\beta)$-noisy, is somewhat unconventional and may lack generality.
3. The experiments do not include comparisons with other learning-augmented algorithms given frequency predictions.
4. A closely related work [1] is not discussed in the paper, where the authors propose learning-augmented search trees that achieve optimality for arbitrary input distributions.

[1] Cao X, Chen J, Chen L, et al. Learning-augmented b-trees[J]. arXiv preprint arXiv:2211.09251, 2022.

**Questions:**

1. In the abstract, the paper claims that for Zipfian distribution, the expected search time for an item is constant, while the traditional skip list or KD tree has an expected search time of $O(\log n)$. However, I could not find a detailed explanation for this claim in the main text. Specifically, is there theoretical support for vanilla skip lists (without learning) showing that the expected search time is $\Omega(\log n)$ under a Zipfian input distribution?
2. In the abstract, the paper claims for arbitrarily incorrect predictions, the algorithm is within a constant factor. Does this exactly come from your $(\alpha,\beta)$-noisy definition?
2. The noise robustness measure is unconventional. I suggest using KL divergence to quantify the distance between the predicted and true frequency distributions, as it may provide a more standard and interpretable measure of prediction accuracy.
3. I suggest adding additional experiment comparisons with other learning-augmented algorithms.

---

> ### Author Response · Authors · 2024-11-20
>
> > The authors slightly overstate their contributions. For example, the claim of constant expected search time under the Zipfian distribution holds only when the exponent $s>1$, which is due to the entropy being of constant order (Lemma 2.3). In other words, for any data structure that achieves optimality has the same property.
>
> > In the abstract, the paper claims that for Zipfian distribution, the expected search time for an item is constant, while the traditional skip list or KD tree has an expected search time of $O(\log n)$. However, I could not find a detailed explanation for this claim in the main text. Specifically, is there theoretical support for vanilla skip lists (without learning) showing that the expected search time is $\Omega(\log n)$ under a Zipfian input distribution?
>
> Thanks for pointing these out. Our intent was not to overstate our contributions and we have clarified in the abstract of the updated version that we achieve constant expected search time for Zipfian distributions with parameter $s>1$.
>
> > Additionally, in the abstract, the claim on the robustness when predictions are arbitrarily incorrect is not what you describe later.
>
> > In the abstract, the paper claims for arbitrarily incorrect predictions, the algorithm is within a constant factor. Does this exactly come from your $(\alpha,\beta)$-noisy definition?
>
> Our exposition mostly focuses on how our guarantees degrade gracefully with the quality of inaccurate predictions through the discussion of $(\alpha,\beta)$ noisy oracles. However, we emphasize that we do actually achieve robustness even when the predictions are arbitrarily incorrect. Both our data structures have depth at most $O(\log n)$ regardless of the quality of the predictions, e.g., our learning-augmented kd tree truncates the estimated probabilities at $\frac{1}{n^2}$. Thus, the expected query time is $O(\log n)$, which is within a constant factor of the query time of standard oblivious search data structures. We have added remarks on this in each section of the main body.
>
> > The noise robustness measure, denoted as $(\alpha,\beta)$-noisy, is somewhat unconventional and may lack generality.
>
> > The noise robustness measure is unconventional. I suggest using KL divergence to quantify the distance between the predicted and true frequency distributions, as it may provide a more standard and interpretable measure of prediction accuracy.
>
> Although $(\alpha,\beta)$-noisy is not necessarily a widely-used notion of robustness, we remark that the same definition was previously introduced and used in [LLW22] and we follow its convention. We agree that other intuitive metrics of robustness, since as total variation distance or KL divergence is an interesting direction for future work.
>
> [LLW22] Honghao Lin, Tian Luo, David P. Woodruff: Learning Augmented Binary Search Trees. ICML 2022: 13431-13440
>
> > The experiments do not include comparisons with other learning-augmented algorithms given frequency predictions.
>
> > I suggest adding additional experiment comparisons with other learning-augmented algorithms.
>
> Thanks for the suggestion, we have performed additional experiments comparing skiplists to balanced binary search trees across various dataset sizes. As expected, the construction time for the skiplists is significantly faster than the construction time for the BSTs across all datasets, due to the latter's necessity of constantly rebalancing the data structure. We have included the relevant discussion and results in the appendix of the updated version of the manuscript.
>
> > A closely related work [1] is not discussed in the paper, where the authors propose learning-augmented search trees that achieve optimality for arbitrary input distributions.
>
> > [1] Cao X, Chen J, Chen L, et al. Learning-augmented b-trees[J]. arXiv preprint arXiv:2211.09251, 2022.
>
> Yes, the work by [1] studies learning-augmented versions of b-trees, which is a different generalization of binary search trees, allowing for nodes with more than two children. They also study the setting where the predictions may be updated, while ultimately still utilizing a data structure that requires rebalancing as data is dynamically changing. Thus, their works have similar drawbacks as binary search trees when compared to skip lists and kd trees.

---

> > ### Comment · Reviewer_Azhq · 2024-11-20
> > **After Rebuttal**
> >
> > I appreciate the authors' thoughtful response to my earlier comments. While I acknowledge the effort made to address my concerns, I still find two points that need further clarification and refinement.
> >
> > I understand the claim that your result achieves $O(1)$ for Zipfian when $s>1$, derived from the fact that the entropy is $O(1)$. However, this result fundamentally comes from the static optimality of your data structure. While it is entirely reasonable to state that "when the prediction is accurate, the data structure achieves static optimality," I believe the emphasis on achieving "constant time" may be overly catchy and potentially misleading to readers unfamiliar with the context. Furthermore, before learning-augmented algorithms, there have been variant of skip lists that achieves optimality [1].
> >
> > Regarding the remark on arbitrary noise, let us consider a specific scenario for clarity. For a Zipfian distribution with $s>1$, the optimal time is indeed $O(1)$. However, if the prediction assigns a uniform probability of $1/n$ to each item, the resulting time complexity becomes $O(\log n)$.
> >
> > [1] Ciriani, Valentina, et al. "Static optimality theorem for external memory string access." The 43rd Annual IEEE Symposium on Foundations of Computer Science, 2002. Proceedings.. IEEE, 2002.

---

> ### Author Response · Authors · 2024-11-20
>
> Thanks for the quick turnaround! We address the two specific remarks below and are happy to answer any further questions.
>
> > I understand the claim that your result achieves $O(1)$ for Zipfian when $s>1$, derived from the fact that the entropy is $O(1)$. However, this result fundamentally comes from the static optimality of your data structure. While it is entirely reasonable to state that "when the prediction is accurate, the data structure achieves static optimality," I believe the emphasis on achieving "constant time" may be overly catchy and potentially misleading to readers unfamiliar with the context. Furthermore, before learning-augmented algorithms, there have been variant of skip lists that achieves optimality [1].
>
> >[1] Ciriani, Valentina, et al. "Static optimality theorem for external memory string access." The 43rd Annual IEEE Symposium on Foundations of Computer Science, 2002. Proceedings.. IEEE, 2002.
>
> We respectfully believe there is some merit in illustrating an example where we can achieve constant time rather than $O(\log n)$ time. However, we understand your point that this has also been done before and may be misleading to unfamiliar readers, and thus we have removed the discussion of Zipfian distributions and constant time from the abstract (and uploaded the new version). We have also incorporated the additional reference [1] among the new related works discussed in the appendix.
>
> > Regarding the remark on arbitrary noise, let us consider a specific scenario for clarity. For a Zipfian distribution with $s>1$, the optimal time is indeed $O(1)$. However, if the prediction assigns a uniform probability of $1/n$ to each item, the resulting time complexity becomes $O(\log n)$.
>
> Since the oracle's prediction is arbitrarily bad, our goal for robustness is to achieve runtime approximately equal to a data structure without access to the oracle (or any auxiliary information). In particular, the main goal of robustness is to ensure that our learning-augmented data structure does not perform *significantly worse* than oblivious data structures, given the erroneous advice. This notion matches the standard definition of robustness in learning-augmented algorithms, e.g., [LLW22].
>
> Moreover, the oblivious data structure cannot know that the query distribution is Zipfian. Thus, even if the distribution is Zipfian and there exists a static tree with runtime $O(1)$, the oblivious binary search tree must be distribution-free, as it does not have access to the sequence of queries. Hence, the oblivious binary search tree has query time $O(\log n)$, which is within a constant-factor of the query time of our learning-augmented skip lists with arbitrarily bad advice, e.g., the uniform distribution you mentioned. Therefore, our algorithms achieve robustness to arbitrarily poor oracles.
>
> [LLW22] Honghao Lin, Tian Luo, David P. Woodruff: Learning Augmented Binary Search Trees. ICML 2022: 13431-13440

---

> > ### Comment · Reviewer_Azhq · 2024-11-24
> > **After Rebuttal and Response**
> >
> > Thank you for the clarification and revision! I have increased the score.

---

### Official Review · Reviewer_btGJ · 2024-11-09

**Soundness:** 2
**Presentation:** 2
**Contribution:** 2
**Rating:** 6
**Confidence:** 3

**Summary:**

The paper considers the problem of augmenting search data structures like skip lists and kd trees with learning advice, in particular, an oracle that gives (possibly erroneous) estimates of the probability of query elements. When the oracle estimate is perfect, the search time for the proposed data structure is within a factor of 2 of the optimal expected search time (upto a constant additive slack). The estimator is also shown to be robust to errors in oracle estimates. For instance, even if the oracle is arbitrarily erroneous, the expected search time is still within a constant factor of the search time for the corresponding oblivious data structure. The superior performance of proposed data structures is also shown with experiments.

Some details of data structure construction:

Skip lists: Skip lists are built in levels, with the bottom level as an ordered linked list. Higher levels store subsets of items to speed up search. Traditional skip lists promote items to higher levels randomly with a fixed probability (e.g., 1/2). In the proposed skip list, if an item's oracle probability exceeds a level dependent threshold, it is promoted up with probability 1 (this leads to faster access for more frequent items). For other items, they are still promoted up with a fixed probability (this leads to robustness to errors in oracle).

Kd tree: In traditional kd trees, each node splits the points according to the median across a selected dimension, iterating between different dimensions at different levels of the tree.  In the proposed data structure, one doesn't necessarily iterate between dimensions. The splitting point and dimension at any node is chosen such that the probability of the query being on either branch is as close to 0.5 as possible.

**Strengths:**

The part on skip lists is a complete and meaningful contribution. The idea of promoting higher frequency elements to higher levels with more probability is both natural and easy to implement. Optimality as well as robustness of the proposed data structure is shown, and it is also shown to be superior in practice with both perfect and erroneous oracles.

**Weaknesses:**

The part on kd trees felt rushed and left me confused about both the motivation and the details of the setting.

Motivation: In the paper, kd trees are considered for doing lookups for high dimensional points (as opposed to nearest neighbor search). But if one is just interested in lookups, it is unclear to me why we need kd trees. Why not just use something like the proposed skip lists after labeling the points from 1 to n? If we want to also support fast membership queries for frequent items not in the dataset, we can also add them to the skip list as is done in the current kd tree construction.

Setting: In the kd tree part, there is a big universe of items and a subset of it constitutes the dataset. The query can be any element in the universe. While in the skip list part, the  the query is always one of the dataset elements. Both settings seem reasonable but I was confused why different settings are considered in the two scenarios. It would be great if this can be clarified in the paper.

Details of the algorithm: It would be good to spend a paragraph discussing various algorithmic choices like was done in the skip list part (space doesn't seem to be a concern as the paper is already half page below the limit). For example, the motivation for handling low probability elements (prob less than 1/n^2) differently was unclear to me as there can be at most n of them. Also, if the authors can elaborate a bit more on how they are handling these points, that would be helpful.

**Questions:**

My main question is about the motivation of the kd tree setting as stated in the weaknesses section.

Another question: Would the proposed skip list data structure be able to handle deletions? For instance, if we delete some elements after constructing the dataset, would it still have the property that for the remaining elements, the data structure is close to optimal wrt the adjusted relative query frequencies? It would be good to include some discussion of deletion in the paper.

Minor formatting comment: The font size used for plots needs to be made larger.

---

> ### Author Response · Authors · 2024-11-20
>
> > The part on kd trees felt rushed and left me confused about both the motivation and the details of the setting.
>
> > My main question is about the motivation of the kd tree setting as stated in the weaknesses section.
>
> > Motivation: In the paper, kd trees are considered for doing lookups for high dimensional points (as opposed to nearest neighbor search). But if one is just interested in lookups, it is unclear to me why we need kd trees. Why not just use something like the proposed skip lists after labeling the points from 1 to n? If we want to also support fast membership queries for frequent items not in the dataset, we can also add them to the skip list as is done in the current kd tree construction.
>
> One major difference between kd trees and skip lists is that the query can be to points that are in the search space, but are not in the dataset. However, the set of queries is generally not uniform and the density of queries can be especially high in some important regions. Thus, it is a natural question whether a data structure that uses machine learning advice to estimate the distribution of queries can achieve better query time for kd trees.
>
> Another more important reason is that kd trees are designed specifically for high-dimensional data, where there is not a natural ordering on the dataset. By partitioning the data recursively along different axes, kd trees can significantly reduce the search space, particularly in moderate-dimensional spaces. Skip lists, on the other hand, are designed more for ordered data and may not be as efficient in handling high-dimensional relationships in a spatial context. Thus, skip lists would require a different structure or additional logic to handle multidimensional point data effectively.
>
> > Setting: In the kd tree part, there is a big universe of items and a subset of it constitutes the dataset. The query can be any element in the universe. While in the skip list part, the the query is always one of the dataset elements. Both settings seem reasonable but I was confused why different settings are considered in the two scenarios. It would be great if this can be clarified in the paper.
>
> Yes, that's a good point. We focus on the setting where queries can be made on the search space rather than the items in the dataset for kd trees, because it can be substantially more problematic if implemented naively, in particular if the number of dimensions $k$ is high, so that even building a balanced tree on the search space could result in prohibitively high query time, as the height of the tree would already be at least $k$.
>
> In fact, our learning-augmented skip lists data structure also generalizes to this setting, provided the oracle is also appropriately adjusted to estimate the query distribution.
>
> We have clarified these points in the updated manuscript.
>
> > Details of the algorithm: It would be good to spend a paragraph discussing various algorithmic choices like was done in the skip list part (space doesn't seem to be a concern as the paper is already half page below the limit). For example, the motivation for handling low probability elements (prob less than 1/n^2) differently was unclear to me as there can be at most n of them. Also, if the authors can elaborate a bit more on how they are handling these points, that would be helpful.
>
> One major difference is that high-dimensional datasets do not have an absolute ordering, and thus we must determine how to "split" the dataset in each iteration, which we do by identifying a dimension with a "good" partition of the dataset. We do this by iterating over the dimensions and picking the dimension with the most balanced split. Another difference is that we must handle both data points and high-frequency queries (though as you mentioned, the corresponding generalization of queries for skiplists would also need to handle this).
>
> There are multiple ways to handle the low-probability elements, since there can be at most $n$ of them, as you noted. Constructing a balanced binary search tree over these elements seemed to be the simplest way to handle them, but for example, we could still build an unbalanced binary search tree where the depth of each item is determined by its predcited query probability.
>
> We have expanded the discussion in the learning-augmented kd trees section to elaborate on these algorithmic choices and in particular, the differences from the learning-augmented skip list.

---

> > ### Author Response · Authors · 2024-11-20
> >
> > > Another question: Would the proposed skip list data structure be able to handle deletions? For instance, if we delete some elements after constructing the dataset, would it still have the property that for the remaining elements, the data structure is close to optimal wrt the adjusted relative query frequencies? It would be good to include some discussion of deletion in the paper.
> >
> > Yes, the simplicity of deletion operations is a major advantage of skip lists! Whenever an item is deleted in the dataset, the corresponding item can be simply be removed from all lists in the skip list. By comparison, the deletion of the item in a balanced binary search tree can often be more nuanced, because though deleting the item from the tree itself is simple, the subsequent rebalancing operation to ensure the tree remains balanced can be more complicated. In fact, the same issue arises for a sequence of insertion operations and we have included not only a discussion of this in the updated version of the paper, but also some empirical evaluations in the supplementary material.
> >
> > > Minor formatting comment: The font size used for plots needs to be made larger.
> >
> > Thanks, we have increased the font size in the updated version. Please let us know if they should be further increased.

---

> > > ### Comment · Reviewer_btGJ · 2024-11-24
> > >
> > > I thank the authors for their response. Just to make sure I understand, one can still use skip lists in the high dimensional setting (along with including high probability non-dataset elements) by converting each point to a scalar (e.g., using row-major order mapping), and get the same guarantees as the proposed kd-trees? But this would result in extremely large numbers when d is large which would be cumbersome to handle, which is where kd-trees can be useful? Or are you saying that it is not possible to get the same guarantees with skip lists in high dim. settings even after converting to a scalar? It would be good to clarify this in the final version of the paper.

---

> > > > ### Author Response · Authors · 2024-11-25
> > > >
> > > > Yes that's correct, it's possible to use skip lists in the high dimensional setting by converting each point to a scalar and get the same guarantees as the proposed kd-trees. However, converting a high-dimensional dataset to a scalar with a total ordering inherently disrupts locality, as the proximity of data points in the original high-dimensional space is not preserved in the scalar representation, leading to poor accuracy in tasks like nearest neighbor search, one of the major applications of kd-trees. We have added this discussion in Section 3 of the updated version of the manuscript, thanks for the suggestion!

---

> > > > > ### Comment · Reviewer_btGJ · 2024-11-25
> > > > >
> > > > > Thank you for the clarification. I have increased my rating.

---

### Author Response · Authors · 2024-11-20

We thank the reviewers for their insightful comments and for the positive feedback provided on the paper, including:
- The part on skip lists is a complete and meaningful contribution. (Reviewer btGJ)
- The idea of promoting higher frequency elements to higher levels with more probability is both natural and easy to implement. (Reviewer btGJ)
- Optimality as well as robustness of the proposed data structure is shown, and it is also shown to be superior in practice with both perfect and erroneous oracles. (Reviewer btGJ)
- Experimental results verify the effectiveness of the proposed algorithms. (Reviewer Azhq)
- The integration of learned frequencies into skip lists and KD trees is well-constructed and achieves optimal performance. (Reviewer Azhq)
- Experiments show that the proposed algorithm outperforms classical algorithms and is robust to noise. (Reviewer Azhq)
- The paper is clearly written and easy to read. The ideas are simple, yet novel and effective. (Reviewer Azhq)
- I appreciate the clear comparison (Reviewer Azhq)
- Provided approach is an elegant way to improve search data structure performance by considering probabilistic nature of the data. (Reviewer i7vt)
- Given the abundance of machine learning and statistical methods to serve as an oracle, proposed approach can be widely used in practice. (Reviewer i7vt)
- New bounds were proven theoretically and experimental results partially support the theoretical findings. (Reviewer i7vt)
- The problems are interesting and relevant, the theoretical results seem solid, and the empirical results show improvement. (Reviewer zTZd)
- It is overall a solid submission. (Reviewer zTZd)

We have uploaded a new version of the manuscript, incorporating reviewer suggestions and addressing the points raised, marking changes in blue text. In particular, we have included:
- An additional experiment comparing binary search trees and skip lists in the appendix.
- An extended discussion on related work in the appendix.
- Additional intuition for the algorithmic design choices for learning-augmented kd-trees.
- Commentary on the robustness of our schemes to arbitrarily bad performance by the oracle.
- Replaced figures to increase the font sizes for improved legibility

We believe these revisions have strengthened the paper and look forward to further feedback. Below, we offer specific responses to the individual comments from each reviewer.

---

### Meta-Review · Area_Chair_Z4sL · 2024-12-20

**Metareview:**

The paper augments two classic data structures, skip lists and kd trees, with advice from an ML oracle. The proposed solution is near-optimal when the oracle is accurate, and is robust to errors of the oracle. The paper provides theoretical guarantees for their approach, and the experiments show its effectiveness. All the reviewers had positive assessments of the paper, and I recommend acceptance.

**Additional Comments On Reviewer Discussion:**

Several questions which the reviewers have been addressed by the authors, and the reviewers have already taken the responses into account in their reviews.

---

### Decision · Program_Chairs · 2025-01-22

Accept (Poster)